# A continuous fish fossil record reveals key insights into adaptive radiation

Nare Ngoepe[1,2 ✉], Moritz Muschick[1,2], Mary A. Kishe[3], Salome Mwaiko[2], Yunuén Temoltzin-Loranca[4], Leighton King[1,2], Colin Courtney Mustaphi[5,6], Oliver Heiri[5], Giulia Wienhues[7], Hendrik Vogel[8], Maria Cuenca-Cambronero[2,9], Willy Tinner[4], Martin Grosjean[7], Blake Matthews[1] & Ole Seehausen[1,2]

Adaptive radiations have been instrumental in generating a considerable amount of life's diversity. Ecological opportunity is thought to be a prerequisite for adaptive radiation[1], but little is known about the relative importance of species' ecological versatility versus effects of arrival order in determining which lineage radiates[2]. Palaeontological records that could help answer this are scarce. In Lake Victoria, a large adaptive radiation of cichlid fishes evolved in an exceptionally short and recent time interval[3]. We present a rich continuous fossil record extracted from a series of long sediment cores along an onshore–offshore gradient. We reconstruct the temporal sequence of events in the assembly of the fish community from thousands of tooth fossils. We reveal arrival order, relative abundance and habitat occupation of all major fish lineages in the system. We show that all major taxa arrived simultaneously as soon as the modern lake began to form. There is no evidence of the radiating haplochromine cichlid lineage arriving before others, nor of their numerical dominance upon colonization; therefore, there is no support for ecological priority effects. However, although many taxa colonized the lake early and several became abundant, only cichlids persisted in the new deep and open-water habitats once these emerged. Because these habitat gradients are also known to have played a major role in speciation, our findings are consistent with the hypothesis that ecological versatility was key to adaptive radiation, not priority by arrival order nor initial numerical dominance.

Adaptive radiations are major components of biological diversity and their study provides insight into the structure and evolutionary dynamics of biodiversity. Adaptive radiation is the diversification of a lineage into an array of species that possess traits enabling them to exploit different environments or resources[4]. It involves a complex set of processes, and identifying the factors (extrinsic and intrinsic) that initiate, facilitate and constrain them remains challenging because of the difficulty of reconstructing conditions at the onset of the classical large radiations. Comparative studies of adaptive radiations provide evidence that ecological opportunity is key to diversification[1,5]. Ecological opportunity refers to the opening up of new niches and the new availability of an abundance of resources. It can trigger diversification through ecological release and ecological expansion, subsequently followed by specialization with niche partitioning during or soon after speciation[6]. Ecological opportunity may lead to a burst of ecological diversification and speciation events and may be triggered in either of several ways including: (1) the colonization of a newly formed environment; (2) the emergence of new resources; (3) the extinction of predators or competitors; and (4) the evolution of a key innovation[7]. All of these factors open up resources that were previously inaccessible to the diversifying lineage.

The ability of colonizing species to be able to access new resources depends on a species' ecological versatility and timing of arrival, relative to other species. Ecologically more versatile taxa can populate a wider range of habitats more quickly and can often access a wider range of dietary resources in individual habitats, which results in increased abundance and wider distribution ranges[8]. Ecological versatility is a prerequisite for rapidly accessing new resources when they first emerge, and thus versatile taxa may experience more immediate opportunities to diversify[9–11]. The other often-invoked but rarely tested hypothesis is that the arrival order of species can determine which one goes on to radiate. This is known as the priority effect, which holds that early colonizing taxa can fully access and subsequently monopolize resources, thereby negatively affecting the opportunities for taxa that arrive later[12,13]. For instance, it has been shown in microbial diversification experiments that early arriving species suppress the diversification

[1]Aquatic Ecology and Evolution, Institute of Ecology and Evolution, University of Bern, Bern, Switzerland. [2]Department of Fish Ecology and Evolution, EAWAG, Swiss Federal Institute for Aquatic Science and Technology, Kastanienbaum, Switzerland. [3]Tanzania Fisheries Research Institute (TAFIRI), Dar es Salaam, Tanzania. [4]Institute of Plant Sciences, University of Bern, Bern, Switzerland. [5]Geoecology, Department of Environmental Sciences, University of Basel, Basel, Switzerland. [6]Center for Water Infrastructure and Sustainable Energy (WISE) Futures, Nelson Mandela African Institution of Science and Technology, Arusha, Tanzania. [7]Institute of Geography & Oeschger Centre for Climate Change Research, University of Bern, Bern, Switzerland. [8]Institute of Geological Sciences & Oeschger Centre for Climate Change Research, University of Bern, Bern, Switzerland. [9]Aquatic Ecology, University of Vic - Central University of Catalonia, Vic, Spain. ✉e-mail: nare.ngoepe@eawag.ch

of late arriving species[14]. However, testing for priority effects and/or differential versatility of colonizing lineages in real adaptive radiations has proved elusive because many of the best-studied adaptive radiations, including most cichlid fish radiations, began millions of years ago[3,7], which makes it almost impossible to assess arrival order or differential ancestral versatility.

The Lake Victoria cichlid fish radiation provides us with unique opportunities in this regard because it unfolded over an exceptionally short and recent geological time interval. Genomic analyses of the Lake Victoria radiation show that much of its genetic diversity stems from a hybridization event between two lineages which had been evolving separately for millions of years[15,16]. Although this event predates the formation of modern Lake Victoria, at approximately 17 thousand years ago (ka) (ref. 17), the genetic variation was present in the founding haplochromine lineage. As this lineage diversified it left a trail of accessible fossils in its wake. The sedimentary record of Lake Victoria provides excellent chronological resolution[17] from which we obtained a rich and continuous fossil record for this study. Previous studies used phylogenetic reconstructions and attributed the rapid evolution of large species diversity in haplochromine cichlids to ecological opportunity (emergence of large deep lakes), a highly evolvable mating trait system and unusual genomic potential due to hybrid ancestry[5,15,16]. However, comparative phylogenetic approaches do not inform us about past community composition of taxa other than the target clade and therefore about relative versatility and arrival order, but such data can be obtained, in principle, from fossils and their abundances and distributions over habitats and through time.

Here, we reconstruct the history of arrival order and the dynamic history of the relative abundances of the radiating taxon and all other major fish taxa across the habitats from the origins of the lake to the present. We took a series of four long composite sediment cores along an onshore-to-offshore transect in the eastern part of Lake Victoria (Methods). First, we asked whether the radiating lineage arrived in the lake before other taxa and whether or not it was numerically dominant over the others. Both early arrival and numerical dominance could afford haplochromine cichlids the opportunity to monopolize resources, permitting them to radiate while potentially inhibiting the radiation of other taxa. Second, we test whether the ancestors of the radiation differed from other contemporaneous taxa in relation to their early habitat utilization, specifically, their ability to persist in new lacustrine habitats that emerged as lake levels rose and that have no analogue in the small streams and swamps from which all colonizing fishes came.

Based on more than 7,500 fish tooth fossils, identified to either family level and/or major lineage within the cichlid family, our evidence suggests that both radiating and non-radiating lineages co-colonized as soon as the lake began to form and the radiating taxon did not dominate the assemblage. However, when the lake became deep and when vast open-water habitats formed, haplochromine cichlids established in offshore deep waters and instantaneously dominated the fish assemblage in these new habitats. In contrast, the hitherto numerically dominant cyprinoids and all other taxa tracked the moving littoral zone but successively abandoned each site as soon as it transitioned to deep water and pelagic habitat. This suggests that cichlids and, in particular, haplochromine cichlids, differed from all other fish that colonized Lake Victoria in their exceptionally large ancestral ecological versatility in relation to habitat. We suggest that the relaxation of selection on traits associated with habitat utilization (that is, ecological opportunity) and the subsequent population expansion into new deep and open-water habitats (that is, ecological release) were crucial for the rapid adaptive radiation of these fishes. Priority effects associated with the order of arrival and relative numerical dominance at the origin of the lake cannot explain which lineages radiated and which did not. In the following, we derive these conclusions in more detail.

## A continuous fish fossil record

We recovered 7,623 fish tooth fossils from four different coring sites together spanning the 17 ka refilling history of Lake Victoria after the late Pleistocene desiccation (Supplementary Table 2 and Supplementary Fig. 1). A total of 931, 1,368, 2,842 and 2,478 fossil teeth were recovered from sieved samples from the coring sites LVC18-S4, LVC18-S1, LVC18-S2 and LVC18-S3 (hereafter referred to as LV4, LV1, LV2 and LV3), from the deepest to the shallowest parts of the lake, respectively (Fig. 1 and Supplementary Table 2). The fossil teeth recovered were generally small, often less than one millimetre in length, but were usually very well preserved (Fig. 3 and Supplementary Fig. 4). Teeth were present in good numbers throughout each of the sediment cores, which permitted the continuous reconstruction of the fish communities in each site (Fig. 1). Although other fish remains such as bones and scales were found too, we present only the fish tooth fossil results here as nearly all teeth can be morphologically assigned to a taxon with certainty (Fig. 3 and Supplementary Fig. 4), something that is not possible with bones or scales. Generally, we observed high fossil abundance in periods of time where a site was a productive wetland or littoral zone, followed by a drop in numbers when the same site became a deep open-lake habitat (Fig. 1). This expected productivity gradient through time repeats itself as a gradient in space across the four cores, where cumulative abundance increases from the offshore to the inshore.

## The fish community assembly

The diagnostic characteristics of the tooth fossils indicated that we recovered representatives of five fish families (Fig. 1): Bagridae, Clariidae, Mochokidae, Cyprinoidea and Cichlidae with haplochromine and oreochromine cichlids diagnosably distinct, which matched the dominant elements of the modern fish community in the lake (Supplementary Table 2). A small proportion of fish teeth (7.33%, $n$ = 562) could not be assigned to any taxon. Bagrid, clariid and mochokid catfish made up 0.12%, 0.13% and 0.24% of the total number of fossil teeth, respectively (Supplementary Table 2). The overall percentage of cyprinoid fossils in the assemblage was 4.53% across the four sites, whereas the entire cichlid family comprised greater than 80% of the assemblage at each site. Across all four sites, the haplochromine cichlids contributed 82.8% (the radiated cichlid lineage) and oreochromine cichlids 4.9% (the non-radiated cichlid lineage) of all the fish fossils found (Supplementary Table 2). This overall composition of these six lineages closely resembles the relative abundance of taxa in contemporary Lake Victoria[18]. However, relative abundances were very different in the first few thousand years of the lake's history. The continuous fossil record demonstrates that haplochromine cichlid fishes, albeit always present in all sites throughout the lake's history, did not come to dominate the assemblage until several thousand years into its history. In fact, cyprinoid fish dominated when the lake first began to fill and for several thousand years afterwards. All other taxa were also present at the origin of the lake but declined later, often below the detection threshold (Fig. 1).

## Habitat transitions and community turnover

The Lake Victoria basin is estimated to be approximately 400,000 years old with a history of drying and refilling, with the last event of complete desiccation in the Last Glacial Maximum (approximately 19–26.5 ka)[19,20] and it started re-wetting about approximately 17 ka (ref. 17). The deepest point of the lake is 68 m and our deepest core LV4 is at 63 m. Our deepest core reveals subaerial exposed deposits or palaeo-Vertisols, possibly with local episodic wetlands, which suggests very dry conditions from at least 20 ka and ending by 16.2 ka at the latest[21]. At about 16 ka, shallow water and wetland conditions established, characterized by abundant *Typha* pollen, which suggests in situ *Typha* growth (greater than 5%), and this condition lasted for approximately 2,000

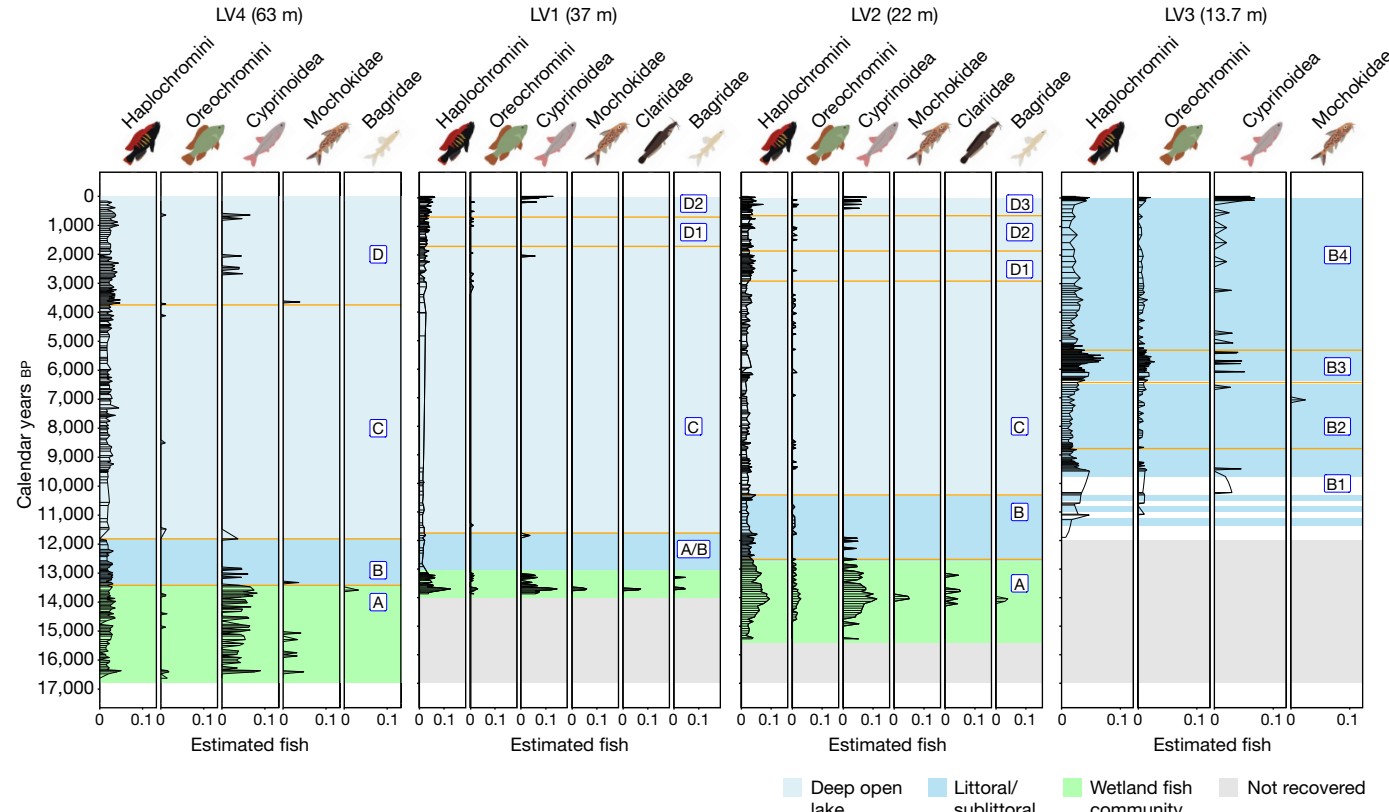

**Fig. 1 | The Lake Victoria fish assemblage since the birth of the modern lake reconstructed from sedimentary deposition of tooth fossils in an onshore–offshore coring transect.** The bars represent the relative abundances of fish of all major taxa derived from fossil influx (fossil concentration/sediment accumulation rates) and are corrected for variation in the average number of teeth among the taxa. The solid orange lines are statistically significant multivariate change points detected across all taxa using the E-divisive method[22]. The letters (A–D) denote different major phases or habitat stages bounded by change points and shared between the sites. Here, A (green shade) represents the wetland fish community phase, B (dark blue) represents the littoral/sublittoral phase, and C and D (light blue) indicate the deep open-lake phases that differ in productivity. The grey shading represents depth not cored and the white spaces in LV3 are intervals not screened for fossils (no data).

years until about 14,000 BP (ref. 17). The topography of Lake Victoria is well characterized and is essentially saucer shaped[20]. From when LV4 began to be inundated, we found continuous traces of fish fossils from the sediment cores and this allowed us to identify major shifts through time in the taxon composition of the Lake Victoria fish community, statistically supported by change point detection analysis[22]. We identified several distinct phases between the change points which we assigned to major habitat and ecosystem stages. Within each site, we compared the abundance of each fish taxon between these stages and inferred the direction of change between the stages using *t*-tests. To be able to convert fossil tooth counts into the relative abundances of fish of the different taxa, we accessed published information and our own computerized tomography scans from preserved fish to obtain information on the number of teeth that typical representatives of each major taxon have (Methods). Four major fish habitat zones are recognized in the modern Lake Victoria: wetlands at the edge of the lake (dominated by emergent hydrophytes, commonly *Cyperus papyrus*); littoral (less than 6 m water depth); sublittoral (6–20 m water depth); and deep waters (greater than 20 m water depth)[23,24].

Palynological and palaeolimnological data for our most offshore site, LV4 (63 m water depth), the first site to be inundated at the onset of the modern lake, suggests that the site was situated in a wetland with low water levels from approximately 16.6 ka until approximately 13.5 ka (ref. 17). An abundance of pollen of semi-aquatic macrophytes and hydrophytes (for example, *Typha* and *Cyperus*) suggests a wetland interspersed with shallow ponds and low water tables until approximately 13.5 ka (ref. 17). Six fish groups are found (Fig. 1): Cyprinoidea was clearly the most abundant group, followed by haplochromine and oreochromine cichlids, and, finally, mochokid, bagrid and clariid catfishes (Fig. 1). This composition resembles that of modern wetlands and shallow lakes across much of eastern and central Africa[25–27]. We refer to this wetland stage, as defined by the fish community, as phase A. By approximately 13 ka *Typha* pollen disappeared and rainforest trees began to spread[17], which suggests an increase in the precipitation/evaporation ratio and site 4 now being distant from wetlands. Because a corresponding change in the pollen records, with the disappearance of *Typha*, was observed simultaneously in site LV2 (ref. 17) which is 40 m shallower than LV4 (LV1 is too short to document this transition), we interpreted it as the signal of our coring sites transitioning to offshore habitat in a deeper lake.

A recent attempt at reconstructing lake levels suggested that LV4 became a deep open-lake habitat just before the change in pollen composition[21]. Around the point in time when the pollen record suggests a transition from a wetland to an offshore lake site for LV4, many fish taxa declined to somewhat lower abundance in the sediment record, including the previously numerous cyprinoids. This change is statistically corroborated by the detection of a first multivariate change point at approximately 13.5 ka (Fig. 1) associated with a significant decrease in cyprinoids but not haplochromine cichlids (Fig. 2). Subsequently, the catfish disappear from the record and the community is composed only of cichlids and cyprinoids. We interpret this community as reflecting a period of lacustrine littoral conditions at the site, which we refer to as phase B. It lasts from approximately 13.5 ka to approximately 11.9 ka, when we detect another significant change point (Fig. 1). At this point in time, most fish taxa declined to below detection level in the sediment record, including the previously numerous cyprinoids.

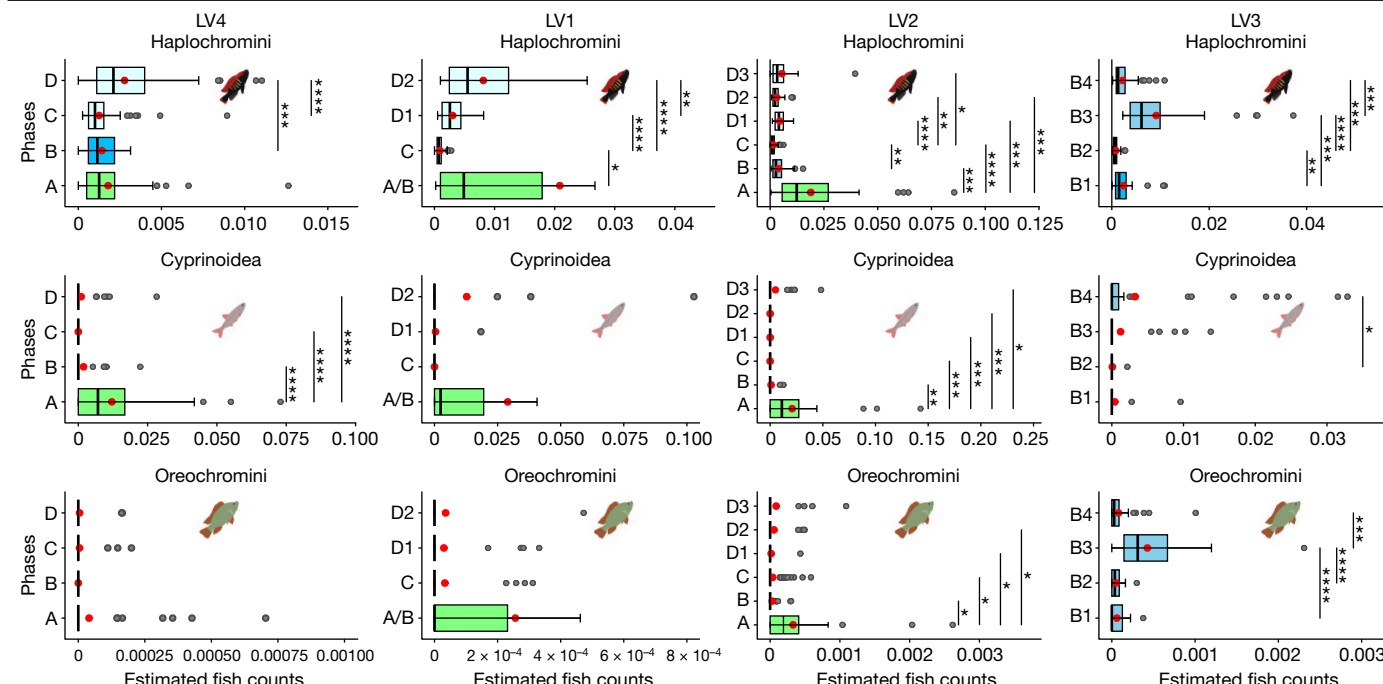

**Fig. 2 | Difference in relative fish abundances between change point delineated habitat phases along the offshore deep (left) to inshore shallow (right) coring transect.** The LV4 site phases were A ($n = 56$), B ($n = 30$), C ($n = 99$) and D ($n = 70$), where $n$ is the number of samples pooled from the change points. The LV1 site phases were A/B ($n = 30$), C ($n = 33$), D1 ($n = 45$) and D2 ($n = 30$). The LV2 phases were A ($n = 40$), B ($n = 38$), C ($n = 121$), D1 ($n = 30$), D2 ($n = 34$) and D3 ($n = 30$). The LV3 site remained throughout in the littoral/sublittoral phases B1 ($n = 30$), B2 ($n = 30$), B3 ($n = 37$) and B4 ($n = 57$). Each plot shows the mean relative abundances of each major fish taxon for each statistically delineated habitat phase and vertical lines indicate significant differences between phases using two-sided $t$-tests. Asterisks indicate the statistical significance (*$P \le 0.05$, **$P \le 0.01$, ***$P \le 0.001$ and ****$P \le 0.0001$) of differences between phases for each taxon. The centre line of the box plot shows the median, the red dot shows the standard error mean, the upper box bound is the 75th percentile and the lower box bound is the 25th percentile. The grey dots show the observations outside the 25th and 75th percentiles. The lower-end line whisker shows the minimum observed value and the upper-end line whisker shows the maximum observed value.

However, while cyprinoids and all taxa other than haplochromines fell below the detection threshold, we consistently recovered haplochromine cichlid teeth from sediments younger than 11.9 ka. This community then resembled the modern (pre-Nile perch invasion) community in offshore Lake Victoria, except that it lacks evidence of the pelagic cyprinoid *Rastrineobola* that is abundant in modern Lake Victoria[28]. We refer to this deep open-lake community as a phase C community, defined by the dominance of haplochromines and nearly complete absence of cyprinoids and catfish.

At the base of our second deepest core, LV1 (37 m water depth), at approximately 13.8 ka, we detected a short interval with a community that resembled the wetland community at the base of LV4 with cyprinoids and haplochromines and the presence of mochokid, bagrid, clariid catfish and oreochromine cichlids (Fig. 1). Whereas cyprinoids had numerically dominated between approximately 16.6 ka and approximately 13.5 ka in LV4, cyprinoids and haplochromines occur in similar numbers at the base of LV1, resembling phase B in LV4. Hence, site LV1 appears to have been a productive wetland for its first 1 ka (approximately 14–13 ka), after which its fish community suggests that the site transitioned to a lake. Just as in LV4, after approximately 13 ka, all taxa abruptly declined below the detection limit except haplochromine cichlids, which declined but remained above the detection limit. This change is reflected by the detection of a multivariate change point at approximately 11.8 ka, which, however, is placed approximately 1 ka after the visually detectable community change (Figs. 1 and 2). For this reason, we assign the section below the change point to a heterogeneous phase within which the wetland transitions to shallow lake littoral (A/B).

Our data from core LV2 (22 m water depth) tell a similar story. This core starts at approximately 15.5 ka and the first 3,000 years has the characteristics of the wetland community described above: co-dominance by cyprinoids and haplochromines and an abundance of oreochromines as well as the regular presence of various catfish. This community persisted from approximately 15.5 ka to approximately 12.6 ka, which is where we detected a multivariate change point (Fig. 1) associated with declining abundances of all taxa (Fig. 2). Subsequently, the catfish disappear from the record and the community is composed only of cichlids and cyprinoids. We interpret this community as reflecting a period of lacustrine littoral conditions at the site, that is, phase B. It lasts from approximately 12.6 ka to approximately 10.4 ka, where we detect another significant change point (Fig. 1). Thereafter, this site (LV2), like LV4 and LV1, loses all fish except haplochromines and occasional oreochromine cichlids. We interpret this as the transition to the deep pelagic lake condition (phase C) and it is marked by a significant change point around approximately 10.4 ka (Fig. 1). This coincides closely with an abrupt drop in hydrophyte pollen (*Cyperus*), an increase in rainforest tree pollen, an abrupt increase in the ratio of rainforest to savanna pollen in LV4 and a similar, albeit less steep, change in LV2 (ref. 17).

Finally, our record for LV3 (13 m water depth), our coring site closest to the shore, starts at approximately 11.5 ka and must have been situated within the littoral and/or sublittoral zone of Lake Victoria (phase B) throughout its history. Consistent with this, we found fossil teeth of haplochromine cichlids, oreochromine cichlids and cyprinoids deposited in the sediment throughout the record. We observe dynamic changes in the abundance of all these taxa, as reflected by several significant multivariate change points (Fig. 1). The dynamic changes are also reflected in significant $t$-tests. We conclude that this site has not had any lasting wetland phase but became first incorporated into the expanding lake as a littoral zone and remained littoral and/or sublittoral throughout the lake's history. Most notably, the abundances of cichlids and cyprinoids increase around 6 ka BP.

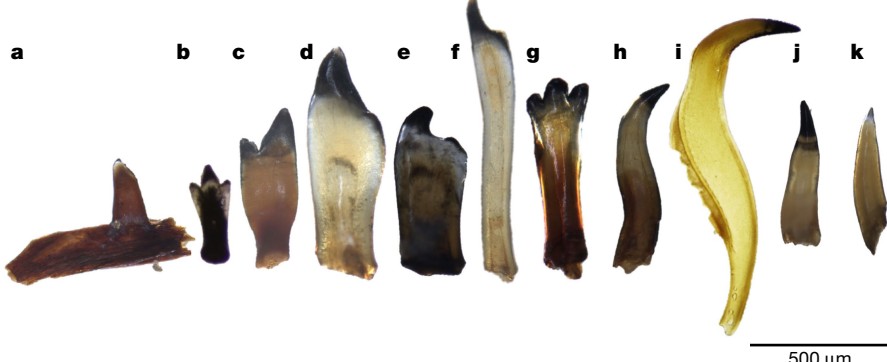

500 μm

**Fig. 3 | Representative fish teeth fossils from multiple taxa as recovered from the sediment cores in Lake Victoria. a**, Cyprinoid pharyngeal tooth attached to the jaw bone. **b**, Haplochromine cichlid oral tooth from an inner tooth row. **c**, Haplochromine cichlid oral tooth from outer tooth row. **d**,**e**, Haplochromine cichlid pharyngeal teeth. **f**, Oreochromine cichlid pharyngeal tooth. **g**, Oreochromine cichlid oral tooth. **h**,**i**, Mochokid catfish teeth from the premaxilla tooth pad on the lower jaw. **j**, Bagrid catfish tooth. **k**, Clariid catfish tooth.

## Cichlids' ecological versatility

Between 13 ka and 12 ka the profundal and pelagic zones (deep and open-water habitats) emerged as totally new habitats that had no analogue in the old stream and wetland landscape that would have constituted fish habitat in the preceding millennia. These new habitats would have potentially provided the ecological opportunity for all fish taxa inhabiting the lake at the time. However, the abundance of all fish except cichlids dropped below the detection threshold as sites transitioned from a wetland and/or shallow lake to the new habitat. This suggests that although various species of fish successfully established in shallow water habitats when the modern lake began to fill, of all the fish in the ecosystem, only cichlids possessed the ecological versatility that allowed them to persist in the new deep and offshore environments. This is consistent with the structure of the modern Lake Victoria fish community where most groups are confined to the near-shore littoral and sublittoral zones, whereas haplochromines occur in large numbers and there are many different species in all habitats across the full depth range and the complete onshore–offshore gradient[28]. It is also consistent with data that show that riverine haplochromines recently introduced to a deep crater lake abound there in deep offshore habitats[29].

Our fossil evidence suggests that the immediate prevalence of haplochromines in the new environments, as they emerged, cannot be a result of evolutionary adaptation during the adaptive radiation process. If this was the case, we would have expected a period of absence of haplochromines followed by their return to the new habitat. The uninterrupted presence of haplochromines, instead, suggests that their ability to persist in the new habitat was already present at the beginning. Clearly, haplochromines, and, to a lesser extent, oreochromines as well, arrived with large habitat versatility that none of the other taxa possessed. Cichlid fishes have been suggested to exhibit unusual ecological versatility with respect to their trophic ecology[30–32], habitat utilization and associated breeding ecology[25] and have highly versatile feeding apparatus[31]. They have been shown to occupy a very wide range of freshwater habitats from little streams and swamps to large and deep lakes[25,33] and a single species can occupy an entire depth range[34]. Our fossil data suggest that haplochromine and oreochromine cichlids persisted in all of our offshore sites when the sites became deep offshore lake habitats, whereas other fish groups disappeared despite them having been numerous at the same sites when they were shallow habitats. We consider this as evidence of the differential habitat versatility. Although oreochromine cichlids displayed ecological versatility too, their numbers decreased to near detection limit in our two most offshore sites, which is consistent with the modern ecology of the two native species *Oreochromis variabilis* and *O. esculentus*, that mainly occurred in the near-shore parts of the lake before their near extirpation[35]. Water depth and onshore–offshore habitat gradients play a fundamental role in haplochromine speciation[34,36,37]. Therefore, it is likely that their large ancestral habitat versatility was key to the rapid emergence of adaptive radiation because it exposed haplochromine populations very quickly to large ecological contrasts to which they would subsequently adapt. The fact that only haplochromines radiated into approximately 500 endemic species, but oreochromines did not, suggests that, although required, versatility alone was likely to be insufficient for adaptive radiation. It is possible that its coincidence with large evolvability, due to the hybrid ancestry[15] of the haplochromine lineage, was key for converting a single ecologically versatile population into many species with distinct ecological specializations.

## Late emergence of pelagic abundance

In all of our offshore cores, we observe a late Holocene increase in the abundance of haplochromine cichlids, followed also by a partial return of cyprinoids in the offshore waters. The multivariate change point analysis places this change in all sites between approximately 3.8 ka and approximately 2.3 ka (approximately 3.8 ka in LV4, approximately 2.3 ka in LV1 and approximately 2.9 ka LV2 (Fig. 1)), but visual inspection of the data is consistent with quasi-simultaneous changes in all three cores around 3.5–3.0 ka (Fig. 1). A significant increase of haplochromines in all three cores was confirmed by *t*-tests. We refer to this late Holocene pelagic abundance phase as phase D. Intriguingly, demographic analyses of whole genome sequence data from modern samples of all haplochromine trophic groups suggest that specialized pelagic planktivores emerged only in the late Holocene (D. Marques et al., manuscript in preparation), which would coincide with the significant increase of haplochromine cichlids in our offshore core sites (Fig. 2). Palynological reconstructions revealed that, at approximately 5 ka, the riparian vegetation changed from rainforest back to savannah with a significant increase in charcoal influx in all three cores[17]. These changes in the terrestrial vegetation cover mark the end of the African humid period[38], reflect long-term changes in rainfall and wind patterns and may have led to increased productivity in the lake. It is noteworthy that, shortly after cichlids increase in the offshore cores, cyprinoids reappear in the record after they had been below the detection threshold for several thousand years. Modern Lake Victoria has one pelagic zooplanktivorous cyprinoid species, *Rastrineobola argentea*. We hypothesize that our fossils represent this species and that its pelagic habits may have emerged around this time, which is something we hope to address with aDNA in the future.

## Conclusions

Our study unravels the interaction and relative importance of key facilitators of adaptive radiation at the onset of a large adaptive radiation. Since haplochromines diversified into hundreds of endemic species in very little time, whereas none of the other lineages did, one might have expected haplochromines to have arrived before the others or to have acquired dominance through abundance before the others. Our evidence suggests that neither of these were the case. We show that when the most offshore coring location became inundated (approximately 17 ka), the fish community immediately resembled modern assemblages of wetlands and rivers in the region in astonishing detail, with an abundance of cyprinoids, haplochromine and oreochromine cichlids and several catfish taxa. This may seem surprising at first, but is not so on closer inspection. Large lakes with endemic species radiations are often likened to offshore islands in the sea. However, contrary to the latter, most large lakes do not start out as geographically isolated from the source region from where they get colonized (that is, the mainland in the case of oceanic islands). In fact, large lakes form in the midst of the 'mainland', which is the river network in the case of freshwater fish. Hence, we should expect many different river fish lineages to colonize newly forming lakes synchronously. This also means that there is little opportunity for priority effects to determine which colonist lineage radiates, and this factor should play less of a role than for radiations on ocean islands. In turn, lineage-specific effects should be more important to explain which lineage radiates in lakes.

The duration of the wetland fish assemblage in Lake Victoria's history closely coincides with a dominance of wetland and savannah vegetation around the lake, based on pollen records[17] and it is lost as soon as the vegetation indicates the emergence of a deep lake. The ability of haplochromine cichlids to fully occupy the new habitats associated with the development of deep lacustrine conditions set them apart from all other colonizing fishes. We propose that this ecological versatility in the new deep open-water habitat has played a decisive role in the diversification process and led to the subsequent ecological dominance of haplochromines. We suggest that the coincidence of ecological opportunity with ecological versatility combined with hybrid ancestry[39] was paramount to setting the stage for the massive and explosive adaptive radiation that occurred in Lake Victoria. Without both opportunity and versatility, the hybrid ancestry of the lineage would have been unlikely to lead to massive diversification[39]. Opportunity and versatility, in turn, would have been unlikely to convert the versatile population into an exceptional radiation, which required large evolvability and hence large genetic variation[39]. Our findings add to understanding the interactions between environmental and lineage-specific factors at the onset of animal adaptive radiations. We suggest that adaptive radiations may be triggered by the coincidence of ecological opportunity with the intrinsic lineage traits of versatility and evolvability.

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

## Methods

### Sediment coring and processing

The sediment cores were collected in 2018 from four sites; LVC18-S1 (located at 1° 6,914′ S, 33° 55,146′ E), LVC18-S2 (located at 1° 7,850′ S, 33° 56,780′ E), LVC18-S3 (located at 1° 6,914′ S, 33° 55,146′ E) and LVC18-S4 (located at 1° 2,966′ S, 33° 47,768′ E). The sites will hereafter be referred to as LV1, LV2, LV3 and LV4, respectively. The cores were collected along a transect of increasing water depths (LV3 at 13 m, LV2 at 22 m, LV1 at 37 m and LV4 at 63 m) and distance from shore (LV3 at 2 km, LV2 at 6 km, LV1 at 9 km and LV4 at 30 km from the shore) in the Shirati Bay area of Lake Victoria (Fig. 1 and Supplementary Fig. 1). This design allowed us to track the productive and most biodiverse littoral zone through time in addition to studying the transition from wetland through the shallow-lake to the deep-lake habitat within sites. The littoral zone would have kept shifting as the lake was refilling and hence studying changes in biodiversity through time in this habitat required coring at multiple localities from offshore to inshore. Cores were taken using an UWITEC platform with metal floats, which was transported to the sites by TAFIRI's RV *Lake Victoria Explorer* and positioned using four large steel anchors. A Niederreiter-type piston-corer with a 3 m drive length, 60 mm liner diameter, motorized hammer and hydraulic core catcher was suspended from the platform. Parallel holes with contiguous drives were cored in sites LV1, LV2 and LV3 and a series of overlapping drives from adjacent holes were used in LV4. Cores were cut into approximately 1-m-long sections, which were sealed and labelled and kept cool during transport and storage. The cores were subsampled contiguously at 2 cm intervals and the samples were then wet sieved through stacked 200 μm and 100 μm mesh-size sieves to retain fish bones, scales and teeth. A ZEISS stereo-microscope Stemi 508 at ×10 magnification (Carl Zeiss) was used to screen and sort subsamples for fossils. The recovered fossils were individually photographed and a detailed record with images and notes was created. Fossils were then visually analysed and compared with the literature and our reference collection to assign each fossil to a taxonomic group. The sediment volume and accumulation rates were adopted from ref. 17.

### The reference collection

A digital photograph catalogue of modern tooth specimens of Lake Victoria fishes was created as a reference collection. The collection comprises 44 species representing seven families that form the modern assemblage in Lake Victoria (Supplementary Table 1). Additionally, published descriptions of teeth together with images and drawings were consulted[23,33,40–43].

### The relative taxon abundance

The relative abundance of each taxon was calculated for each sampling interval using the average of the typical number of teeth a fish has in each taxon based on published tooth counts of representative species of each major taxon and our reference collection. For example, *Astatotilapia nubila*, the ancestral type haplochromine cichlid that occupies the wetlands around Lake Victoria[44], has between 138 and 147 teeth in the oral and pharyngeal jaws combined[45]. The oreochromine cichlids have between 60 and 1,000 teeth in the oral and pharyngeal jaws combined[35,46]. Cyprinoid fish lack teeth in the oral jaws and have only 11–15 teeth in the lower pharyngeal jaws and no teeth in the upper pharyngeal jaws[47,48]. For catfish families, Mochokidae have 6–11 teeth[49], Bagridae have about 50 teeth[41] and Clariidae have about 20 teeth[50]. The implication from the neontological data is that it is twice as likely to recover a tooth from an individual oreochromine than from an individual haplochromine, ten times less likely to recover a tooth from an individual cyprinoid and mochokid and five times less likely to recover a tooth from an individual clariid catfish than from a haplochromine. The fish fossil influx was obtained from fossil concentration per area per year (the concentration of fossils/area per year) (Supplementary Fig. 3) and we observed an even depositing of fish teeth, which suggests that multiple teeth were unlikely to stem from the same individual.

**Statistical analysis.** For detecting significant distributional changes through time in the composition and abundance of taxa, we used the multivariate change points detection E-divisive method from the ecp R package v.3.1.3 (ref. 22). The method detects significant changes in mean and variance using the hierarchical divisive estimation algorithm of a time series. Initially, all the observations were included in a single cluster. Then, at each step, the algorithm split the data into hierarchal homogeneous clusters. Our dataset was analysed and visualized using Rstudio v.4.3.1 (ref. 51) with packages rstatix v.0.7.2, ggplot2 v.3.4.2, tidypaleo v.0.1.3, patchwork 1.1.2, scales v.1.2.1, ggtext v.0.1.2 and dplyr v.1.1.2. We created the box plots to summarize each taxon's mean relative abundance for each habitat phase. A two-sided *t*-test was computed to infer significant differences between phases.

### Sediment geochronology, vegetation and lake level dynamics

The lake level changes were inferred from ref. 21 and the chronology and age models were based on ref. 17. The chronology was based on a total of 93 samples of terrestrial macrofossils that were radiocarbon dated using the MICADAS accelerator mass spectrometry system at the Laboratory for the Analysis of Radiocarbon with AMS at the University of Bern[52]. The terrestrial macrofossil ages were used to construct three independent models with 95% (2σ) probabilities using rbacon[53] in R software and the IntCal20 calibration curve[54]. The biostratigraphy of each site was used to cross check the resulting ages[17]. The age model for LV3 was based on ¹⁴C dates from a total of 26 samples (23 of charcoal, 3 bulk sediment) following the same approach as in ref. 17.

### Reporting summary

Further information on research design is available in the Nature Portfolio Reporting Summary linked to this article.

## Data availability

All the data generated and analysed in the current study, including the code to process the data and reproduce all the figures presented here, are available at figshare repositories (https://doi.org/10.6084/m9.figshare.23895783, https://doi.org/10.6084/m9.figshare.23895771, https://doi.org/10.6084/m9.figshare.23828475).

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

**Acknowledgements** We thank all collaborators who supported the field expedition to collect the sediment cores. The coring team members included P. Boltshauser-Kaltenrieder, W. Tanner, S. Brügger, A. Bolland and members of the TAFIRI team. The coring platform crew consisted of E. Sombe (boat captain), H. Kalima (engineer), B. Jumbe (crew), T. Mohammed (technician), D. Bwathondi (crew), D. Chacha (crew) and H. Ramadhani (cook). Coring was undertaken with the permission of COSTECH, research permit number 2018-237-NA-2018-57, and was supported by a University of Bern faculty strategy grant to O.S., A. Sapfo Malaspinas, W.T., O.H., M.G. and F. Anselmetti, and by the Institute of Ecology and Evolution and the Institute of Plant Sciences of the University of Bern. We thank E. Jemmi, A. Boila, R. Lazaro and A. Viertler for their technical support in reference collection expansion, fossil screening and sediment sieving. This research was supported by SNSF Sinergia grant CRSII5_183566, which was funded by the Swiss National Science Foundation and awarded to O.S., M.G., T. Gilbert and B.M.

**Author contributions** The study design and data interpretation were carried out by N.N., O.S. and M.M. The coring expedition was led by M.M., S.M. and M.A.K. The subsampling of the sediment cores and sieving was conducted by Y.T.-L. and N.N. N.N. screened for fossils, analysed the data and conceptualized the study with support from M.M. and O.S. N.N. and O.S. wrote the manuscript with contributions from all authors. All authors are part of Sinergia consortium, except S.M.

**Competing interests** The authors declare no competing interests.

**Additional information**
**Correspondence and requests for materials** should be addressed to Nare Ngoepe.

# Reporting Summary

## Statistics

For all statistical analyses, confirm that the following items are present in the figure legend, table legend, main text, or Methods section.

| n/a | Confirmed | |
|---|---|---|
| ☐ | ☒ | The exact sample size (*n*) for each experimental group/condition, given as a discrete number and unit of measurement |
| ☐ | ☒ | A statement on whether measurements were taken from distinct samples or whether the same sample was measured repeatedly |
| ☐ | ☒ | The statistical test(s) used AND whether they are one- or two-sided<br>*Only common tests should be described solely by name; describe more complex techniques in the Methods section.* |
| ☐ | ☒ | A description of all covariates tested |
| ☐ | ☒ | A description of any assumptions or corrections, such as tests of normality and adjustment for multiple comparisons |
| ☐ | ☒ | A full description of the statistical parameters including central tendency (e.g. means) or other basic estimates (e.g. regression coefficient) AND variation (e.g. standard deviation) or associated estimates of uncertainty (e.g. confidence intervals) |
| ☐ | ☒ | For null hypothesis testing, the test statistic (e.g. *F*, *t*, *r*) with confidence intervals, effect sizes, degrees of freedom and *P* value noted<br>*Give P values as exact values whenever suitable.* |
| ☒ | ☐ | For Bayesian analysis, information on the choice of priors and Markov chain Monte Carlo settings |
| ☐ | ☒ | For hierarchical and complex designs, identification of the appropriate level for tests and full reporting of outcomes |
| ☒ | ☐ | Estimates of effect sizes (e.g. Cohen's *d*, Pearson's *r*), indicating how they were calculated |

*Our web collection on statistics for biologists contains articles on many of the points above.*

## Software and code

Policy information about availability of computer code

| Data collection | We did not use any software to collect data. |
|---|---|
| Data analysis | We used Rstudio version 4.3.1 with packages rstatix v0.7.2, ggplot2 v3.4.2, tidypaleo v0.1.3, patchwork 1.1.2, scales v1.2.1, ggtext v0.1.2, rbacon v3.1.1, and dplyr v1.1.2. |

For manuscripts utilizing custom algorithms or software that are central to the research but not yet described in published literature, software must be made available to editors and reviewers. We strongly encourage code deposition in a community repository (e.g. GitHub). See the Nature Portfolio guidelines for submitting code & software for further information.

## Data

Policy information about availability of data

All manuscripts must include a data availability statement. This statement should provide the following information, where applicable:
- Accession codes, unique identifiers, or web links for publicly available datasets
- A description of any restrictions on data availability
- For clinical datasets or third party data, please ensure that the statement adheres to our policy

All the data generated and analyzed in the current study, including the code to process the data and reproduce all the figures presented here are available on figshare repository.

# Research involving human participants, their data, or biological material

Policy information about studies with [human participants or human data](). See also policy information about [sex, gender (identity/presentation), and sexual orientation]() and [race, ethnicity and racism]().

| | |
|---|---|
| Reporting on sex and gender | NA |
| Reporting on race, ethnicity, or other socially relevant groupings | NA |
| Population characteristics | NA |
| Recruitment | NA |
| Ethics oversight | NA |

Note that full information on the approval of the study protocol must also be provided in the manuscript.

# Field-specific reporting

Please select the one below that is the best fit for your research. If you are not sure, read the appropriate sections before making your selection.

☐ Life sciences ☐ Behavioural & social sciences ☒ Ecological, evolutionary & environmental sciences

For a reference copy of the document with all sections, see [nature.com/documents/nr-reporting-summary-flat.pdf]()

# Life sciences study design

All studies must disclose on these points even when the disclosure is negative.

| | |
|---|---|
| Sample size | 7623 fish teeth fossils |
| Data exclusions | fossil bones, scales and unassignable fossils |
| Replication | NA |
| Randomization | NA |
| Blinding | NA |

# Behavioural & social sciences study design

All studies must disclose on these points even when the disclosure is negative.

| | |
|---|---|
| Study description | NA |
| Research sample | NA |
| Sampling strategy | NA |
| Data collection | NA |
| Timing | NA |
| Data exclusions | NA |
| Non-participation | NA |
| Randomization | NA |

# Ecological, evolutionary & environmental sciences study design

All studies must disclose on these points even when the disclosure is negative.

| | |
|---|---|
| Study description | We used 7623 fish teeth fossils from multiple sites in Lake Victoria to address some major hypothesis at the beginning of an adaptive radiation. We reveal arrival order, relative abundance and habitat occupation of all major fish lineages in the system. We calculated fish estimates from a combination of fish fossil influx for each taxon and the average of the typical number of teeth the modern fish has in the mouth deduced from the published tooth counts of representative species of each major taxon and our reference collection. We reported the fish estimates and detected significant distributional changes through time in the fish composition and abundance of taxa with the multivariate changepoints detection E-divisive method. |
| Research sample | Multiple representing a wide range of present day fish families found in Lake Victoria were used in the study to interpret the fossil data and assign to taxa. From Alestidae (Brycinus jacksonii), Bagridae (Bagrus docmak), Cichlidae (Astatoreochromis alluaudi, Astatotilapia nubila, Enterochromis paropius, Gaurochromis hiatus, Haplochromis purple yellow, Harpagochromis cf. serranus, Labrochromis stone, Lipochromis melanopterus, Lithochromis sp. (scraper pseudonigricans), Lithochromis sp. yellow chin pseudonigricans, Mbipia lutea, Mbipia mbipi, Neochromis gigas, Neochromis omniceruleus, Neochromis rufocaudalis, Neochromis sp. (uniscupid scraper), Paralabidochromis chilotes, Paralabidochromis cyaneus, Paralabidochromis flavus, Paralabidochromis sp. rockkribensis, Paralabidochromis sp. (short snout scraper), Platytaeniodus degeni, Psammochromis riponianus, Ptyochromis sauvagei or P. fisheri, Ptyochromis xenognathus, Pundamilia macrocephala, Pundamilia nyererei, Pundamilia pundamilia, Pundamilia sp. (pink anal), Yssichromis laparogramma, Yssichromis pyrrhocephalus), Clariidae (Clarias sp.), Cypriniformes (Rastrineobola argentea, Labeo victorianus, Enteromius sp.), Latidae (Lates niloticus), Mochokidae (Synodontis victoriae), Mastacembelidae (Mastacembelus frenatus), and Oreochromini (Oreochromis variabilis, Oreochromis esculentus, Oreochromis niloticus, and Oreochromis leucosticus). And the sediment core dates and volumes are from Temoltzin-Loranca et al. 2023. |
| Sampling strategy | The sediment cores were collected in 2018 from four sites; LVC18-S1 (located at 01°06,914' S, 33°55,146' E), LVC18-S2 (located at 01°07,850' S, 33°56,780' E), LVC18-S3 (located at 01°06,914' S, 33°55,146' E) and LVC18-S4 (located at 01°02,966' S, 33°47,768' E). We refer to the sites as LV1, LV2, LV3 and LV4, respectively. The cores were collected along a transect of increasing water depths (LV3 at 13 m, LV2 at 22 m, LV1 at 37 m, LV4 at 63 m) and distance from shore (LV3 at 2 km, LV2 at 6 km, LV1 at 9 km, LV4 at 30 km from the shore) in the Shirati Bay area of Lake Victoria. |
| Data collection | The sediment cores were taken by a team led by Moritz Muschick, Mary Kishe and Salome Mwaiko, using an UWITEC platform with metal floats, which was transported to the sites by TAFIRI's RV Lake Victoria Explorer and positioned using four large steel anchors. A Niederreiter-type piston-corer with a 3m drive length, 60mm liner diameter, motorized hammer and hydraulic core catcher was suspended from the platform. Parallel holes with contiguous drives were cored in sites LV1, LV2 and LV3, and a series of overlapping drives from adjacent holes in LV4. Cores were cut into ~1m long sections, which were sealed and labelled and kept cool during transport and storage. Yunuen Temoltzin-Loranca sub-sampled the cores contiguously at 2 cm intervals, and the samples were then wet-sieved through stacked 200 µm and 100µm mesh-size sieves to retain fish bones, scales and teeth. Nare Ngoepe screened the 3nature portfolio | reporting summary March 2021 samples with ZEISS stereo-microscope Stemi 508 at 10x magnification (Carl Zeiss, Heidelberg, Germany) sorted the subsamples for fossils. The recovered fossils were individually photographed and a detailed record with images and notes was created. Fossils were then visually analyzed and compared with literature and our reference collection to assign each fossil to a taxonomic group. |
| Timing and spatial scale | The sediments cores were collected from 10.10.2018 to 31.10.2018. The sub-sampling, fossil screening, fossil sorting and taxa assignments were continuous from June 2019 to December 2022. |
| Data exclusions | Fossil bones and scales were excluded as they cannot be morphologically assigned to a taxon, that information we could only obtain from fish teeth fossils. |
| Reproducibility | All the calculations on our data can be reproduced. The study design is also reproduceable and we provided a detailed reference images of teeth that can also be used. |
| Randomization | We had digital photograph catalogue of modern tooth specimens of Lake Victoria fishes that we used as a reference to assign to different taxa. The collection comprised 40 species representing seven families that form the modern assemblage in Lake Victoria. We also used published descriptions of teeth together with images and drawings to assign the fossil data into taxa. |
| Blinding | No data blinding was performed, all the fish taxon had unique teeth morphologies. |

Did the study involve field work? ☒ Yes ☐ No

# Field work, collection and transport

| | |
|---|---|
| Field conditions | Core collection was undertaken in October 2018 on the central western shore of Lake Victoria at the locations given below. Calm winds in the early morning hours allowed coring from an anchored platform in both shallow and deep waters. The joint expedition of the University of Bern and the Tanzania Fisheries Research Institute (TAFIRI) was carried out using the modular UWITEC coring platform of the Institute of Plant Sciences of the University of Bern and TAFIRI's research vessel Lake Victoria Explorer. |
| Location | Tanzania, Lake Victoria, Shirati Bay, Four sites; LVC18-S1 (located at 01°06,914' S, 33°55,146' E), LVC18-S2 (located at 01°07,850' S, 33°56,780' E), LVC18-S3 (located at 01°06,914' S, 33°55,146' E) and LVC18-S4 (located at 01°02,966' S, 33°47,768' E). |

| Access & import/export | Coring was undertaken with permission of COSTECH under research permit No. 2018-237-NA-2018-57. The coring platform and ancillary research equipment was temporarily imported and samples exported with permission of the Ministry of Livestock and Fisheries, United Republic of Tanzania. |
|---|---|
| Disturbance | Assembly, transport and positioning of the platform, and coring of 73.8 m (total) of sediment cores caused only very localised disturbance to the sediment surface. No tracer fluids were used. |

# Reporting for specific materials, systems and methods

We require information from authors about some types of materials, experimental systems and methods used in many studies. Here, indicate whether each material, system or method listed is relevant to your study. If you are not sure if a list item applies to your research, read the appropriate section before selecting a response.

## Materials & experimental systems

| n/a | Involved in the study |
|---|---|
| ☒ ☐ | Antibodies |
| ☒ ☐ | Eukaryotic cell lines |
| ☒ ☐ | Palaeontology and archaeology |
| ☒ ☐ | Animals and other organisms |
| ☒ ☐ | Clinical data |
| ☒ ☐ | Dual use research of concern |
| ☒ ☐ | Plants |

## Methods

| n/a | Involved in the study |
|---|---|
| ☒ ☐ | ChIP-seq |
| ☒ ☐ | Flow cytometry |
| ☒ ☐ | MRI-based neuroimaging |

## Antibodies

| Antibodies used | NA |
|---|---|
| Validation | NA |

## Eukaryotic cell lines

Policy information about cell lines and Sex and Gender in Research

| Cell line source(s) | NA |
|---|---|
| Authentication | NA |
| Mycoplasma contamination | NA |
| Commonly misidentified lines (See ICLAC register) | NA |

## Palaeontology and Archaeology

| Specimen provenance | NA |
|---|---|
| Specimen deposition | NA |
| Dating methods | NA |

☐ Tick this box to confirm that the raw and calibrated dates are available in the paper or in Supplementary Information.

| Ethics oversight | NA |
|---|---|

Note that full information on the approval of the study protocol must also be provided in the manuscript.

## Animals and other research organisms

Policy information about studies involving animals; ARRIVE guidelines recommended for reporting animal research, and Sex and Gender in Research

| Laboratory animals | NA |
|---|---|

| Wild animals | NA |
|---|---|
| Reporting on sex | NA |
| Field-collected samples | NA |
| Ethics oversight | NA |

Note that full information on the approval of the study protocol must also be provided in the manuscript.

# Clinical data

Policy information about clinical studies

All manuscripts should comply with the ICMJE guidelines for publication of clinical research and a completed CONSORT checklist must be included with all submissions.

| Clinical trial registration | NA |
|---|---|
| Study protocol | NA |
| Data collection | NA |
| Outcomes | NA |

# Dual use research of concern

Policy information about dual use research of concern

## Hazards

Could the accidental, deliberate or reckless misuse of agents or technologies generated in the work, or the application of information presented in the manuscript, pose a threat to:

No | Yes
☒ | ☐ Public health
☒ | ☐ National security
☒ | ☐ Crops and/or livestock
☒ | ☐ Ecosystems
☒ | ☐ Any other significant area

## Experiments of concern

Does the work involve any of these experiments of concern:

No | Yes
☒ | ☐ Demonstrate how to render a vaccine ineffective
☒ | ☐ Confer resistance to therapeutically useful antibiotics or antiviral agents
☒ | ☐ Enhance the virulence of a pathogen or render a nonpathogen virulent
☒ | ☐ Increase transmissibility of a pathogen
☒ | ☐ Alter the host range of a pathogen
☒ | ☐ Enable evasion of diagnostic/detection modalities
☒ | ☐ Enable the weaponization of a biological agent or toxin
☒ | ☐ Any other potentially harmful combination of experiments and agents

# Plants

| Seed stocks | NA |
|---|---|
| Novel plant genotypes | NA |
| Authentication | NA |

# ChIP-seq

## Data deposition

☐ Confirm that both raw and final processed data have been deposited in a public database such as GEO.

☐ Confirm that you have deposited or provided access to graph files (e.g. BED files) for the called peaks.

Data access links
*May remain private before publication.*
> NA

Files in database submission
> NA

Genome browser session
(e.g. UCSC)
> NA

## Methodology

Replicates
> NA

Sequencing depth
> NA

Antibodies
> NA

Peak calling parameters
> NA

Data quality
> NA

Software
> NA

# Flow Cytometry

## Plots

Confirm that:

☐ The axis labels state the marker and fluorochrome used (e.g. CD4-FITC).

☐ The axis scales are clearly visible. Include numbers along axes only for bottom left plot of group (a 'group' is an analysis of identical markers).

☐ All plots are contour plots with outliers or pseudocolor plots.

☐ A numerical value for number of cells or percentage (with statistics) is provided.

## Methodology

Sample preparation
> NA

Instrument
> NA

Software
> NA

Cell population abundance
> NA

Gating strategy
> NA

☐ Tick this box to confirm that a figure exemplifying the gating strategy is provided in the Supplementary Information.

# Magnetic resonance imaging

## Experimental design

Design type
> NA

Design specifications
> NA

Behavioral performance measures
> NA

## Acquisition

| | |
|---|---|
| Imaging type(s) | NA |
| Field strength | NA |
| Sequence & imaging parameters | NA |
| Area of acquisition | NA |

Diffusion MRI ☐ Used ☐ Not used

## Preprocessing

| | |
|---|---|
| Preprocessing software | NA |
| Normalization | NA |
| Normalization template | NA |
| Noise and artifact removal | NA |
| Volume censoring | NA |

## Statistical modeling & inference

| | |
|---|---|
| Model type and settings | NA |
| Effect(s) tested | NA |

Specify type of analysis: ☐ Whole brain ☐ ROI-based ☐ Both

| | |
|---|---|
| Statistic type for inference | NA |

(See Eklund et al. 2016)

| | |
|---|---|
| Correction | NA |

## Models & analysis

| n/a | Involved in the study |
|---|---|
| ☒ | ☐ Functional and/or effective connectivity |
| ☒ | ☐ Graph analysis |
| ☒ | ☐ Multivariate modeling or predictive analysis |

| | |
|---|---|
| Functional and/or effective connectivity | NA |
| Graph analysis | NA |
| Multivariate modeling and predictive analysis | NA |

