## [Peer Review File · Nature]

Manuscript Title: A continuous 17ka fish fossil record from Lake Victoria reveals key insights into the beginnings of adaptive radiation

Reviewer Comments & Author Rebuttals

Reviewer Reports on the Initial Version:

Referees' comments:

Referee #1 (Remarks to the Author):

The Rift Lake cichlid radiations—and particularly those of Lake Victoria—are renowned for their evolutionary radiation. This study provides a fossil perspective on the temporal pattern of diversification, data which are novel not only for this radiation—but as the authors point out—for just about any radiation that has been studied with modern molecular systematic approaches. As someone who has followed the burgeoning knowledge of the cichlid radiation for years, I found this to be a fascinating new twist, a nice addition to an already spectacular case study of evolutionary diversification.

The data—to this non-fish-specialist—are compelling that there is no evidence for priority effects operating at the authors' study sites. As the authors point out, this is a viable hypothesis to explain why some lineages radiate at a particular place and others don't, a question of great importance with little resolution for most radiations. This is an unusual attempt to test this hypothesis with actual historical data (as opposed to inference from phylogenetic analysis, or just hand waving).

I do have two qualms about the paper:

First, having rejected the priority-effect hypothesis, the authors argue that it is the extreme ecological versatility of cichlids that accounts for their exceptional radiation compared to other lineages. There are no data from this study that test this hypothesis: the conclusion rests, rather, on a variety of other lines of evidence that argue for such versatility, as well as the plausibility of the explanation given the chronological sequence. I don't think that it is possible with the authors' data set to test this hypothesis explicitly; I only point this out to say that the major advance of this paper is the test of the priority effect, and the conclusions about the role of versatility are more inferred than demonstrated.

Second, knowing little of Lake Victoria, it occurred to me that there is an alternative explanation for the rapid dominance of cichlids in deep water habitats once those habitats emerge. The authors seem to be suggesting that this occupation occurred as a result of the fish already present at that site moving into the deep-water habitats as they develop (or persisting into them as the habitat changes). Another possibility, of course, is that the cichlids arrived from some other part of the lake where deeper water habitats had appeared earlier. If possible, this would seem to complicate the authors' scenario—maybe cichlids evolved their deep-water adaptations elsewhere, rather than being predisposed to be able to use such habitats. My interpretation of supplementary figure 1 suggests that this is a possibility. I suspect that the authors have a response to this point, and I suggest they present it in the paper (and apologies if I missed it).

Referee #2 (Remarks to the Author):

The manuscript by Ngoepe et al. describes a unique study of the fish assemblages of Lake Victoria using fossil teeth in sediment cores. The Lake Victoria haplochromine cichlid fish system is of fundamental global importance for our understanding of evolutionary process. It is estimated that over 500 species have evolved from common ancestry within the remarkably short timescale since the modern lake formed, some 17000 years ago. The genetic diversity that contributed to this radiation, however, is much older, and previous work by some members the team has shown that the Lake Victoria haplochromine radiation may have originated from genetic variation that was seeded by a hybridization event between at least two distinct evolutionary lineages of riverine cichlids.

For this study, the authors drilled four cores along an inshore – offshore gradient. Sections of the cores were dated and classified into phases – namely wetland, littoral/sublittoral and deep open lake. Fossil teeth in each section of the cores were classified as belonging to one of several major lineages of fishes in the region – namely Haplochromini, Oreochromini, Cyprinidae, Mochokidae, Clariidae and Bagridae.

The results show that between 17000 and 13000 years ago, the lake was shallow wetland - and all major fish species groups were present, but when the deep lake formed the open waters were dominated by almost exclusively by haplochromines. This pattern is compatible with the hypothesis that only haplochromines had the capability to occupy the novel deep water habitat, while other species groups were restricted to littoral/sublittoral or wetland habitats. This central result is important because it demonstrates the haplochromine diversification was not a consequence of their earlier arrival in the lake than other lineages. Instead it suggests that haplochromines have a predisposition to diversification not possessed by other lineages.

The study is of considerable significance because it is the first, to my knowledge, to comprehensively describe the changes in a fish community during the process of adaptive radiation. It clearly demonstrates that the temporal order of colonisation is not the key determinant of whether radiation takes place in cichlid fishes. The results neatly correspond to evidence that haplochromines are typically the fish lineage that is most prone to diversification in other East African lakes that are <1 million years old - including Lakes Malawi, Rukwa, Masoko and Chala. The key claim of the authors, that arrival order is not the key determinant of whether diversification takes place, is supported by the evidence.

My primary suggestions relate to how additional context would improve the clarity and repeatability of the work. I also outline an additional set of more minor issues to consider that will help improve the readability and accuracy.

- 1) The paper would benefit from a more comprehensive introduction to the known geology of the basin of the lake basin. It would be valuable to additionally relate the current study to the earlier ages of the formation of Lake Victoria basin (~400,000 years), and the evidence of the repeated filling and desiccation of the lake basin – as described by Johnson et al. (2000; Ambio).
- 2) The approach to dating the sediment layers containing the fossils is critical to this paper, but the treatment of this issue is restricted to references to a recently published paper. Ideally the paper would explain the methods used to constrain the dates, while also directing the reader to a detailed explanation in the published paper.
- 3) The methods used to identify the fossils are touched upon, and one or two examples from each major lineage are shown, However, I wonder if the illustrations represent relatively pristine teeth with little taphonomic change. Using the evidence and information provided in the paper I question whether the assignment of fish fossils to the taxonomic groups would be externally repeatable. Ideally a more comprehensive reference image library would be made available in the supporting information/data for the paper, alongside greater range of example fossil teeth assigned to each taxon. There should be a more comprehensive description of the key diagnostic traits of tooth

traits of each taxon to support the assignment, ideally refining/replacing the text lines 523-628.

Minor comments

Line 9. A "bout of rapid diversification" is not a definition of an adaptive radiation, so this would be better reworded.

Line 14 Consider "a large adaptive radiation of cichlid fishes".

Line 20. Here and elsewhere it may be better to refer to the filling of the "modern lake".

Line 28. Consider "Adaptive radiations are major components of biological diversity".

Line 30 Consider "possess traits enabling them to exploit".

Line 33 Consider "at their onset".

Line 34 Consider "studies of adaptive radiations".

Line 36. Ecological opportunity is simply the opening up of novel niches and does not make specific predictions about abundance of within niches resources and/or the strength of selection.

Line 66. It is worth being more precise here, in that the modern species diversity most likely arose in the last 17000 years, although the genetic diversity the radiation contains arose earlier.

Line 134 – "relative abundance of these six lineages in contemporary"

Line 162. It seems strange here to start referring to cyprinids as "carp-like fish" when you have already mentioned cyprinids in text, tables and figures earlier.

Line 248. This line about the dominance not evolving would be better if reworded.

Line 298. I am indeed not particularly surprised that newly formed wetlands are rapidly colonised by cyprinids, haplochromines, oreochromines and clariid catfishes. This process has taken place multiple times following the apparent complete desiccation of Lake Chilwa in Malawi (1910s, 1960s and 1990s). Presumably the recolonisation in both systems was enabled by relict populations, perhaps in shallow higher altitude streams. While the Chilwa droughts have been short-lived and nowhere near as extensive as the 14-18 ka megadrought that desiccated Lake Victoria, the evidence that these same species groups rapidly expanded in abundance following lake refilling in both systems is interesting in this context.

Table 1. This would benefit from revision/correction to make it more accurate and precise to the study system.

Table 1. Taxonomic names: Here you refer to *Barbus*, in SI Table 1 *Enteromius* but not *Barbus*. It would be good to check for consistency in the generic names of the non-cichlid taxa.

You provide two examples of *Clarias* with different habitat preferences, but it is not clear why you chose these specific taxa. Are these exemplar taxa? It should be explained in the legend.

Table 1. The general habitat preferences of some lineages are not entirely correct. Bagridae, as a family, can be found in rivers, shallow lakes and across depth zones in deep lakes. *Bagrus docmak*, the LV species, according to Yongo and Agembe (2021), "inhabits lakes, swamps and rivers in both shallow and deep waters associated with rocky bottoms and coarse substrates (Lock, 1982)." Mochokidae are also found in deep lakes, and in rivers.

Line 400 Consider "mochokid"

There does not seem to be any reference in the manuscript or other documentation about where the data is/will be stored and made available.

Supplementary Table 1. Some informal names are in italics.

Page 22-23. The information on the different groups is inconsistent in quality and quantity, and ideally would provide only the key information for the paper. Some information is peripheral to the study - for example on broader biogeographic patterns. I suggest reconsidering this section to systematically provide information on 1) the number of species known from the catchment, 2) the range of habitats that they presently occupy, and 3) the key distinguishing features of the dentition that has enabled you to assign identities to each of the fossil teeth. This should be critical section of diagnostic text to support the taxonomic assignments.

Line 590. *Clarias liocephalus* is not endemic to Lake Victoria. I am not sure what is being referred to by "Clarias Xenoclaris" – is it *Xenoclaris eupogon*? While the latter may have been confined to deeper waters, *Clarias liocephalus* is found in very shallow wetland habitats.

Line 593. There is no mention of *Enteromius* here, yet these are often the most abundant cyprinids in shallow water habitats.

Line 607. Consider being more precise in this "swamp worm"?

Line 612. *Synodontis afrofisheri* is not endemic to the Lake Victoria catchment.

Line 620 Consider "herbivorous and are occasionally piscivorous"

General comment. *Pseudocrenilabrus* does not seem to have been considered, yet may have been an abundant taxon in palaeo-wetland habitats.

Referee #3 (Remarks to the Author):

A. Summary of the key results

The MS reports the details of a painstaking and innovative survey of thousands of fish tooth fossils from 4 sediment cores from the bottom of Lake Victoria, covering a period of >17,000 years which includes the restoration of the lake following an extensive period of drought. The authors take the opportunity to use this dataset to test and answer a very important question, namely whether colonisation sequence might have determined which lineages underwent adaptive radiation and which did not. The data clearly discount this explanation at least for several of the major lineages in Lake Victoria, with 5 non-radiating lineages present from the outset alongside the hyper-radiating haplochromine cichlids. Indeed, it provides strong evidence that that haplochromines were not even the dominant lineage in shallow-water wetland conditions, only really coming to dominate in deep water habitats, once they became established in the lake.

B. Originality and significance: if not novel, please include reference

The results are novel and significant. A nice commonsense explanation is given about the way that big lakes differ from islands (crater lakes maybe less so) in appearing in the middle of 'freshwater mainlands' in the form of rivers systems and so their fauna is likely to be highly diverse from the outset with essentially many simultaneous colonisers. I think this will help to persuade a general readership of the plausibility and likely generality of the results.

C. Data & methodology: validity of approach, quality of data, quality of presentation

The paper is extremely well-written and presented and the analyses and interpretations look sound to me.

D. Appropriate use of statistics and treatment of uncertainties

The data analysis looks appropriate and E-divisive method looks very powerful and seems to yield

a reasonable level of congruence between cores, given how much noise there is likely to be in the data for environmental and sampling reasons

E. Conclusions: robustness, validity, reliability

Obviously, this is a single lake and a subset of the possible lineages, but getting this data is obviously a huge undertaking and the lineages available seem the best candidates - Oreochromini radiate in Lake Natron, Mochokidae in Lake Tanganyika, Clariidae in Lake Malawi and Cypriniformes are globally the dominant freshwater fishes and have radiated in Lake Tana- although none of these have formed huge radiations like the haplochromines have in Lake Malawi and Victoria. Overall, I think the work will lead to further 'replications' in other lacustrine systems, facilitating the testing of the generality of the findings.

F. Suggested improvements: experiments, data for possible revision

I suggest the authors consider the following minor comments: (i) The taxonomic treatment of the cyprinids should be updated to reflect current classifications. Rastrineobola is now in the Danionidae, but Cypriniformes (order) or Cyprinoidea (Superfamily) can still be used to unite this with taxa currently placed in Cyprinidae. Barbus (Table 1) is no longer applied to African taxa which are now Enteromius (as in Supplementary Table 1) and Labeobarbus. (ii) I am glad to see Astatotilapia nubilata referred to as the 'ancestor' of the haplochromine radiation and urge the authors to resist pressure from other reviewers/editors to try to shoehorn it into cladistic terminology which probably doesn't reflect reality! (iii) It is slightly confusing that the numbering of the cores goes from shallowest to deepest in the order 3, 2, 1, 4. It might be easier to follow if it went 1, 2, 3, 4 or 4, 3, 2, 1. (iv) Supplementary Table 1: Oreochromini is a tribe of cichlid fishes (alongside haplochromini) and not a separate family from Cichlidae and Oreochromis variabilis should be under Oreochromini not Mastacembelidae. (v) Figure 1 is generally nice, but the details of all the images of tiny fish in the lake are presumably meant to show lots of cichlids in deep water and a diversity of lineages in the shallows: it might work better to use fewer, bigger images- and perhaps omit the left hand side of the image.

G. References: appropriate credit to previous work?

Appropriate credit has been given to relevant work, as far as I know.

H. Clarity and context: lucidity of abstract/summary, appropriateness of abstract, introduction and conclusion

Presentation & writing generally seem fine (as above).

Author Rebuttals to Initial Comments:

Reviewer 1:

The Rift Lake cichlid radiations “and particularly those of Lake Victoria” are renowned for their evolutionary radiation. This study provides a fossil perspective on the temporal pattern of diversification, data which are novel not only for this radiation (but as the authors point out) for just about any radiation that has been studied with modern molecular systematic approaches. As someone who has followed the burgeoning knowledge of the cichlid radiation for years, I found this to be a fascinating new twist, a nice addition to an already spectacular case study of evolutionary diversification.

We are very pleased to see reviewer 1 sharing our enthusiasm about the novelty of our work.

The data “to this non-fish-specialist” are compelling that there is no evidence for priority effects operating at the authors’ study sites. As the authors point out, this is a viable hypothesis to explain why some lineages radiate at a particular place and others don’t, a question of great importance with little resolution for most radiations. This is an unusual attempt to test this hypothesis with actual historical data (as opposed to inference from phylogenetic analysis, or just hand waving).

We are happy to see that reviewer 1 considers our results compelling evidence that there is no evidence for priority effects.

I do have two qualms about the paper:

Comment: *First, having rejected the priority-effect hypothesis, the authors argue that it is the extreme ecological versatility of cichlids that accounts for their exceptional radiation compared to other lineages. There are no data from this study that test this hypothesis: the*

conclusion rests, rather, on a variety of other lines of evidence that argue for such versatility, as well as the plausibility of the explanation given the chronological sequence. I don't think that it is possible with the authors data set to test this hypothesis explicitly; I only point this out to say that the major advance of this paper is the test of the priority effect, and the conclusions about the role of versatility are more inferred than demonstrated.

Response: Indeed, our data provide evidence that priority effects did not facilitate the cichlid radiation in Lake Victoria. The hypothesis of differential ecological versatility of haplochromine cichlids is based on the large literature on cichlid ecology and life history traits. Haplochromines are versatile and opportunistic feeders ((Liem and Osse, 1975; Bouton, Seehausen and Van Alphen, 1997) and many others) with a highly versatile feeding apparatus (Liem and Osse, 1975). They also occupy a very wide range of freshwater habitats from little streams and swamps to large and deep lakes (Fryer and Iles, 1972; Lowe-McConnell, 1987). In lakes, they are known to occupy the entire oxygenated water depth range. Whereas after they radiated in a lake, the different species of haplochromines have become specialised on different parts of the total habitable depth range, in smaller and younger lakes where radiations have not (yet) occurred, a single species typically occupies the entire depth range (Malinsky *et al.*, 2015). Remarkably, the associated habitat expansion from shallow streams to great depths of lakes can happen within decades after a river haplochromine has been introduced into a deep lake (Moser *et al.*, 2018). This suggests extensive habitat versatility within a single population. Key to understanding this habitat versatility of haplochromines is probably their reproductive system. Haplochromines are female mouthbrooders that carry their eggs in the mouth from the moment of egg-laying until the young fish become independent. Haplochromines share this mode of reproduction with oreochromine cichlids, whereas none of the other fish in the Lake Victoria system possess this mode. Because females continuously clean and move the eggs in their mouth, this allows them to breed in habitats most other fish cannot breed either because the substrate is unsuitable to attach eggs (i.e. soft substrate) or because oxygen concentrations are too low for eggs to develop. Both are typically the case in deep and open waters of lakes. It is for this reason that Rosemary Lowe-McConnell has suggested mouthbrooding to be a life-history key innovation that predisposes haplochromine and oreochromine cichlids to undergo adaptive radiation in lakes (Lowe-McConnell, 1987). The hypothesis makes the prediction that haplochromines and oreochromines are more successful than most other fish groups in colonising and establishing in the offshore and deep sectors of lakes. This is a prediction that with our fossil data we can test for the first time for any cichlid radiation. Consistent with the

prediction, our data suggest that haplochromine and oreochromine cichlids persisted in all our offshore sites when the sites became deep offshore lake habitats, whereas other fish groups disappeared despite them having been numerous at the same sites when they were shallow habitats. We consider this support for the differential habitat versatility hypothesis. We have now tried to clarify this further in the text (lines 266 to 275).

Comment: *Second, knowing little of Lake Victoria, it occurred to me that there is an alternative explanation for the rapid dominance of cichlids in deep water habitats once those habitats emerge. The authors seem to be suggesting that this occupation occurred as a result of the fish already present at that site moving into the deep-water habitats as they develop (or persisting into them as the habitat changes). Another possibility, of course, is that the cichlids arrived from some other part of the lake where deeper water habitats had appeared earlier. If possible, this would seem to complicate the authors scenario. maybe cichlids evolved their deep-water adaptations elsewhere, rather than being predisposed to be able to use such habitats. My interpretation of supplementary figure 1 suggests that this is a possibility. I suspect that the authors have a response to this point, and I suggest they present it in the paper (and apologies if I missed it).*

Response: We thank the reviewer for this important question. We would like to clarify here why it is essentially impossible for the cichlids to have arrived from deepwater habitat. Lake Victoria had been completely desiccated during the arid periods in the Last Glacial Maximum (~19–26.5 ka) (Johnson *et al.*, 1996; Tryon *et al.*, 2016) and recent evidence shows it started refilling about ~ 17k years ago (Temoltzin-Loranca *et al.*, 2023). The deepest point of the lake is 68m and our deepest core LV4 is at 63m. Our deepest core (LV4) reveals subaerial exposed deposits or paleo-Vertisols, possibly with local episodic wetlands, suggesting very dry conditions from at least 20ka and ending by 16.2ka at the latest. At about 16 ka, shallow water and wetland conditions establish, characterised by abundant *Typha* pollen suggesting in-situ *Typha* growth (>5%, Temoltzin-Loranca *et al.*, 2023). This condition lasts for approximately two thousand years until about 14ka (Temoltzin *et al.* 2023). This is the phase in which we recover a wetland fish community from our fossils. Also, the topography of Lake Victoria is well-characterised and is essentially saucer-shaped (Johnson *et al.*, 1996). There are no deep sectors anywhere in the lake and there are no deep lakes anywhere else in the catchment either. It is hence extremely unlikely that fishes could have colonised our sites from deep water. Given the topography, when our offshore site LV4 began to become inundated, there cannot have

been habitat deeper than 5m anywhere else in the catchment, and this must have been so for several thousand years. We have explained this more fully in the manuscript text now (lines 147 to 156).

Reviewer 2:

The manuscript by Ngoepe et al. describes a unique study of the fish assemblages of Lake Victoria using fossil teeth in sediment cores. The Lake Victoria haplochromine cichlid fish system is of fundamental global importance for our understanding of evolutionary process. It is estimated that over 500 species have evolved from common ancestry within the remarkably short timescale since the modern lake formed, some 17000 years ago. The genetic diversity that contributed to this radiation, however, is much older, and previous work by some members the team has shown that the Lake Victoria haplochromine radiation may have originated from genetic variation that was seeded by a hybridization event between at least two distinct evolutionary lineages of riverine cichlids.

For this study, the authors drilled four cores along an inshore-offshore gradient. Sections of the cores were dated and classified into phases; namely wetland, littoral/sublittoral and deep open lake. Fossil teeth in each section of the cores were classified as belonging to one of several major lineages of fishes in the region; namely Haplochromini, Oreochromini, Cyprinidae, Mochokidae, Clariidae and Bagridae.

The results show that between 17000 and 13000 years ago, the lake was shallow wetland - and all major fish species groups were present, but when the deep lake formed the open waters were dominated by almost exclusively by haplochromines. This pattern is compatible with the hypothesis that only haplochromines had the capability to occupy the novel deep water habitat, while other species groups were restricted to littoral/sublittoral or wetland habitats. This central result is important because it demonstrates the haplochromine diversification was not a consequence of their earlier arrival in the lake than other lineages. Instead it suggests that haplochromines have a predisposition to diversification not possessed by other lineages.

The study is of considerable significance because it is the first, to my knowledge, to comprehensively describe the changes in a fish community during the process of adaptive radiation. It clearly demonstrates that the temporal order of colonisation is not the key determinant of whether radiation takes place in cichlid fishes. The results neatly correspond

to evidence that haplochromines are typically the fish lineage that is most prone to diversification in other East African lakes that are <1 million years old - including Lakes Malawi, Rukwa, Masoko and Chala. The key claim of the authors, that arrival order is not the key determinant of whether diversification takes place, is supported by the evidence.

We thank the reviewer for their positive assessment and we are happy to see that the reviewer agrees that our data demonstrates the haplochromine diversification was not a consequence of an earlier arrival in the lake compared to other lineages, but that it suggests instead that haplochromines have a predisposition to diversification not possessed by other lineages.

My primary suggestions relate to how additional context would improve the clarity and repeatability of the work. I also outline an additional set of more minor issues to consider that will help improve the readability and accuracy.

Comment: *1) The paper would benefit from a more comprehensive introduction to the known geology of the basin of the lake basin. It would be valuable to additionally relate the current study to the earlier ages of the formation of Lake Victoria basin (~400,000 years), and the evidence of the repeated filling and desiccation of the lake basin as described by Johnson et al. (2000; Ambio).*

Response: We thank the reviewer for this suggestion and we revised and added what is known about the history of the lake basin (line 147 to 149) **“The Lake Victoria basin is estimated to be ~400 000 years old with a history of repeated drying and refilling, with the last event of complete desiccation in the Last Glacial Maximum (~19–26.5 ka) (Johnson et al., 1996; Tryon et al., 2016) after which it began to wetten again about ~ 17k years ago (Temoltzin-Loranca et al., 2023).”**

Comment: *2) The approach to dating the sediment layers containing the fossils is critical to this paper, but the treatment of this issue is restricted to references to a recently published paper. Ideally the paper would explain the methods used to constrain the dates, while also directing the reader to a detailed explanation in the published paper.*

Response: We have now added more details of the chronology in the methods section (line 447 to 456) **“The lake level changes were inferred from (Wienhues, et al in submitted. 2023),**

and the chronology and age models were based on (Temoltzin-Loranca *et al.*, 2023). This chronology is based on a total of 93 samples of terrestrial macrofossils that were radiocarbon dated using the MICADAS accelerator mass spectrometry (AMS) system at the Laboratory for the Analysis of Radiocarbon with AMS (LARA) at the University of Bern (Szidat *et al.*, 2014). The terrestrial macrofossil ages were used to construct three independent models (for LV4, LV1 and LV2) with 95% (2σ) probabilities using rbacon (Blaauw and Andrés Christen, 2011) in R software and the IntCal20 calibration curve (Reimer *et al.*, 2020). The biostratigraphy of each site was used to crosscheck the resulting ages (Temoltzin-Loranca *et al.*, 2023). The age model for LV3 was constructed based on ^{14}C dates from a total of 26 samples (23 of charcoal, 3 bulk sediment) following the same approach as (Temoltzin-Loranca *et al.*, 2023).”

Comment: 3) *The methods used to identify the fossils are touched upon, and one or two examples from each major lineage are shown, However, I wonder if the illustrations represent relatively pristine teeth with little taphonomic change. Using the evidence and information provided in the paper I question whether the assignment of fish fossils to the taxonomic groups would be externally repeatable. Ideally a more comprehensive reference image library would be made available in the supporting information/data for the paper, alongside greater range of example fossil teeth assigned to each taxon. There should be a more comprehensive description of the key diagnostic traits of tooth traits of each taxon to support the assignment, ideally refining/replacing the text lines 523-628.*

Response: We thank the reviewer for this suggestion. Although we have not entirely replaced lines 523-628, we have refined them and also included as supplementary material a comprehensive catalogue of images from our tooth reference collection. Additionally, we are now showing a much greater range of examples of fossil teeth assigned to each taxon from the four cored sites and also included teeth that exhibit notable diagenetic damage. (line 614 - 685)

Minor comments

Comment: *Line 9. About of rapid diversification is not a definition of an adaptive radiation, so this would be better reworded.*

Response: We have adopted the suggestion by the reviewer and now reworded our line to “Adaptive radiations have been instrumental in generating a considerable amount of life's diversity.” (line 9 to 10)

Comment: Line 14 Consider "a large adaptive radiation of cichlid fishes"

Response: we have adopted the suggestion by the reviewer and added “a” and it now reads as: “In Africa’s Lake Victoria, a large adaptive radiation of cichlid fishes evolved in an exceptionally short and recent time interval.” (line 14)

Comment: Line 20. Here and elsewhere it may be better to refer to the filling of the "modern lake"

Response: we have added “modern”; “We show that all major taxa arrived simultaneously as soon as the modern lake began to form.” (line 20 and line line 251)

Comment: Line 28. Consider "Adaptive radiations are major components of biological diversity"

Response: we have adopted the suggestion by the reviewer and added “components” and and it now reads as: “Adaptive radiations are a major components of biological diversity, and their study provides insight into the structure and evolutionary dynamics of biodiversity.” (line 28)

Comment: Line 30 Consider "possess traits enabling them to exploit"

Response: Adopted the suggestion by the reviewer and added “enabling them” and it now reads as: “Adaptive radiation is the rapid diversification of a lineage into an array of species that possess traits enabling them to exploit different environments or resources” (line 30)

Comment: Line 33 Consider "at their onset"

Response: the sentence did not read well when we changed “the onset” to “at their onset”

Comment: Line 34 Consider "studies of adaptive radiations"

Response: Adopted the suggestion by the reviewer and added “s” and it now reads as:

“Comparative studies of adaptive radiations provide evidence that ecological opportunity is key to diversification” (line 34)

Comment: Line 36. Ecological opportunity is simply the opening up of novel niches and does not make specific predictions about abundance of within niches resources and/or the strength of selection.

Response: we have adopted the suggestion by the reviewer and reworded the sentence to “Ecological opportunity refers to the **opening up of novel niches** and becoming available of an abundance of resources.” (line 35-36)

Comment: Line 66. It is worth being more precise here, in that the modern species diversity most likely arose in the last 17000 years, although the genetic diversity the radiation contains arose earlier.

Response: We agree. Much of the genetic diversity is much older than any of the species in this radiation. Many of the genetic polymorphisms arose from hybridization between two several million years old species that happened prior to the formation of the lakes in the region (Meier et al. 2019; McGee et al 2020). We have added this information “**Genomic analyses of the LV radiation show that much of its genetic diversity stems from a hybridization event between two lineages which had been evolving separately for millions of years** (Meier et al., 2017; McGee et al., 2020). **Although this event pre-dates the formation of modern LV ~17ka** (Temoltzin-Loranca et al., 2023) **the genetic variation was present in the founding haplochromine lineage. As this lineage diversified** it left a trail of accessible fossils in its wake.”. (line 65 to 68)

Comment: Line 134 "relative abundance of these six lineages in contemporary"

Response: we have adopted the suggestion by the reviewer and reworded the sentence to “This overall composition **of these six lineages** closely resembles the relative abundance of taxa in contemporary Lake Victoria” (line 137-138)

Comment: Line 162. It seems strange here to start referring to cyprinids as "carp-like fish" when you have already mentioned cyprinids in text, tables and figures earlier.

Response: we have adopted the suggestion by the reviewer and removed "carp-like fish" (line 174)

Comment: Line 248. This line about the dominance not evolving would be better if reworded.

Response: we have adopted the suggestion by the reviewer and clarified it to “Our fossil evidence suggests that the immediate prevalence of haplochromines in the new environments, as they emerged, cannot be a result of evolutionary adaptation during the adaptive radiation process. If this was the case, we would have expected a period of absence of haplochromines followed by their return to the new habitat. The uninterrupted presence of haplochromines, instead, suggests that their ability to persist in the new habitat was already present right from the beginning.” (line 260 - 265)

Comment: Line 298. I am indeed not particularly surprised that newly formed wetlands are rapidly colonised by cyprinids, haplochromines, oreochromines and clariid catfishes. This process has taken place multiple times following the apparent complete desiccation of Lake Chilwa in Malawi (1910s, 1960s and 1990s). Presumably the recolonisation in both systems was enabled by relict populations, perhaps in shallow higher altitude streams. While the Chilwa droughts have been short-lived and nowhere near as extensive as the 14-18 ka megadrought that desiccated Lake Victoria, the evidence that these same species groups rapidly expanded in abundance following lake refilling in both systems is interesting in this context.

Response: We thank the reviewer for this interesting addition. It is indeed very interesting that the repeated drying and refilling of Lake Chilwa is associated with rapid recolonization by the same set of fish taxa that we infer to have been the first to recolonize the emerging modern Lake Victoria. We are delighted to include the reference to the Lake Chilwa situation and Lake Liambezi. Indeed, although quite shallow (ranging from 3 to 5m water depth) and occasionally drying up, upon lake level fluctuations and recolonization Lake Chilwa fish assemblage included haplochromines, oreochromines, small barbs and clariid catfishes (Njaya *et al.*, 2011). Similarly, records for the ephemeral Lake Liambezi in Namibia show the fish assemblage that colonised included cyprinids and catfishes such as clariids in the colonizing taxa that began in 2007 (Peel *et al.*, 2019).

Comment: Table 1. This would benefit from revision/correction to make it more accurate and precise to the study system. Table 1. Taxonomic names: Here you refer to *Barbus*, in SI Table 1 *Enteromius* but not *Barbus*. It would be good to check for consistency in the generic names of the non-cichlid taxa.

Response: we have revised Table 1 and changed *Barbus* to *Enteromius*

Comment: You provide two examples of *Clarias* with different habitat preferences, but it is not clear why you chose these specific taxa. Are these exemplar taxa? It should be explained in the legend.

Response: We have revised Table 1 and updated the habitat preferences to cover the Clariidae family instead of only the two taxa in the previous version.

Comment: Table 1. The general habitat preferences of some lineages are not entirely correct. *Bagridae*, as a family, can be found in rivers, shallow lakes and across depth zones in deep lakes. *Bagrus docmak*, the LV species, according to Yongo and Agembe (2021), "inhabits lakes, swamps and rivers in both shallow and deep waters associated with rocky bottoms and coarse substrates (Lock, 1982)." *Mochokidae* are also found in deep lakes, and in rivers.

Response: We have looked closely into the reviewer's suggestions and though we agree, we highlight some lake-specific attributes of the LV basin. For example, *Bagrus* is indeed associated with both deep and shallow rock habitats (Graham, 1929). The rocky habitats play a huge role in breeding where these catfish attach their eggs to the rocks and the juveniles prefer exposed rocky shores where they feed on lithophilic insects (Corbet, 1960). But since LV is a flat saucer-shaped lake (Johnson *et al.*, 1996), the floor in deeper and offshore habitats is covered with soft sediment, leaving no exposed rock substrate that would support the breeding and juveniles of *Bagrus*. Rocky habitats in LV are the shores of the mainland and of some offshore islands. It seems hence relatively unlikely that *Bagrus* form populations in the deep and offshore waters.

With regard to *Synodontis* (Mochokidae), these indeed inhabit deeper waters in Lake Tanganyika. But there they have evolved a very interesting form of brood parasitism by which catfish lay their eggs into the nest of mouthbrooding cichlids as the latter spawn (Sato, 1986).

The catfish eggs and larvae then benefit from the same advantages of mouthbrooding care that cichlid eggs and larvae enjoy, and that permits them to reproduce at water depth where low oxygen concentrations are limiting the reproduction of other fish (Day and Wilkinson, 2006). However, the Lake Victoria species of Mochokidae did not evolve such relationships with cichlids and they seem to prefer shallow areas less than 20m deep (*Synodontis afrofisheri*) and 14m deep (*Synodontis victoriae*) (Witte, De Winter and Van Densen, 1995) and tend to migrate into rivers to spawn. We have edited Table 1 to reflect these habitat preferences in the context of Lake Victoria.

Comment: Line 400 Consider "mochokid"

Response: we have changed from Mochokid to "mochokid" (line 434)

Comment: *There does not seem to be any reference in the manuscript or other documentation about where the data is/will be stored and made available.*

Response: The data will be provided as a spreadsheet, together with the rscript used to analyze and plot our data.

Comment: *Supplementary Table 1. Some informal names are in italics.*

Response: We have updated Table 1 and removed italics on the informal names and included them in quotations“ ”

Comment: *Page 22-23. The information on the different groups is inconsistent in quality and quantity, and ideally would provide only the key information for the paper. Some information is peripheral to the study - for example on broader biogeographic patterns. I suggest reconsidering this section to systematically provide information on 1) the number of species known from the catchment, 2) the range of habitats that they presently occupy, and 3) the key distinguishing features of the dentition that has enabled you to assign identities to each of the fossil teeth. This should be critical section of diagnostic text to support the taxonomic assignments.*

Response: The information on different groups is amended to include only the key and relevant

details 1) the number of species occurring in Lake Victoria, 2) the range of habitat they occupy, and 3) the distinguishing features of the dentition. (line 620-677)

Comment: Line 590. *Clarias liocephalus* is not endemic to Lake Victoria. I am not sure what is being referred to by "*Clarias Xenoclaris*" is it *Xenoclaris eupogon*? While the latter may have been confined to deeper waters, *Clarias liocephalus* is found in very shallow wetland habitats.

Response: Amended "There are six species of clariids occurring in Lake Victoria. They are found in deeper waters (*Xenoclaris eupogon*), shallow wetland habitats (*Clarias liocephalus*), inshore waters including marginal waterlily and papyrus swamps (*Clarias gariepinus*)" (line 628-631)

Comment: Line 593. There is no mention of *Enteromius* here, yet these are often the most abundant cyprinids in shallow water habitats.

Response: Amended "In LV, they are found in a variety of habitats including inshore waters and migrate to rivers to spawn (genera *Labeo* and *Labeobarbus*), and swamps, streams, weed beds, shallow rocky habitats (various species in the genus *Enteromius*), exposed shores and the uppermost littoral zone (genus *Garra*), the pelagic (genus *Rastrineobola* and *Enteromius profundus*)" (line 641 - 645)

Comment: Line 607. Consider being more precise in this "swamp worm"

Response: As the reviewer suggested, we only retained the relevant information and as a result "swamp worm" was cut out

Comment: Line 612. *Synodontis afrofisheri* is not endemic to the Lake Victoria catchment.

Response: This is true, we now only report the species occurring in Lake Victoria "The family of African catfishes has two species occurring in Lake Victoria (*Synodontis afrofisheri* and *Synodontis victoriae*)". (line 634 - 636)

Comment: Line 620 Consider "herbivorous and are occasionally piscivorous"

Response: As the reviewer suggested, we only retained the relevant information and diet preference was no longer included in our text.

Comment: General comment. Pseudocrenilabrus does not seem to have been considered, yet may have been an abundant taxon in palaeo-wetland habitats.

Response: We did consider *Pseudocrenilabrus* and we have a CT-scanned specimen that formed part of our reference (as shown in supplementary figure 3). The phenotypic diversity we recovered from the fossil record leaves no doubt that the assemblage we recovered was one of *Astatotilapia*-derived teeth, resembling *A. nubila* and modern radiation haplochromines although we cannot rule out that some *Pseudocrenilabrus* were present. In fact, we would expect them to have been present in the wetland assemblages.

Reviewer 3:

A. Summary of the key results

The MS reports the details of a painstaking and innovative survey of thousands of fish tooth fossils from 4 sediment cores from the bottom of Lake Victoria, covering a period of >17,000 years which includes the restoration of the lake following an extensive period of drought. The authors take the opportunity to use this dataset to test and answer a very important question, namely whether colonisation sequence might have determined which lineages underwent adaptive radiation and which did not. The data clearly discount this explanation at least for several of the major lineages in Lake Victoria, with 5 non-radiating lineages present from the outset alongside the hyper-radiating haplochromine cichlids. Indeed, it provides strong evidence that that haplochromines were not even the dominant lineage in shallow-water wetland conditions, only really coming to dominate in deep water habitats, once they became established in the lake.

We thank reviewer 3 for their positive evaluation. We are glad they agree with our conclusion to clearly discount the colonisation sequence as an explanation for which lineage radiated and which did not.

B. Originality and significance: if not novel, please include reference

The results are novel and significant. A nice commonsense explanation is given about the way that big lakes differ from islands (crater lakes maybe less so) in appearing in the middle of “freshwater mainlands” in the form of rivers systems and so their fauna is likely to be highly diverse from the outset with essentially many simultaneous colonisers. I think this will help to persuade a general readership of the plausibility and likely generality of the results.

We are glad to hear reviewer 3 shares our perspective on how lakes are expected to get colonized and our excitement about the novelty and significance of our findings.

C. Data & methodology: validity of approach, quality of data, quality of presentation

The paper is extremely well-written and presented and the analyses and interpretations look sound to me.

We thank reviewer 3 for their positive evaluation.

D. Appropriate use of statistics and treatment of uncertainties

The data analysis looks appropriate and E-divisive method looks very powerful and seems to yield a reasonable level of congruence between cores, given how much noise there is likely to be in the data for environmental and sampling reasons

E. Conclusions: robustness, validity, reliability

Obviously, this is a single lake and a subset of the possible lineages, but getting this data is obviously a huge undertaking and the lineages available seem the best candidates - Oreochromini radiate in Lake Natron, Mochockidae in Lake Tanganyika, Clariidae in Lake Malawi and Cypriniformes are globally the dominant freshwater fishes and have radiated in Lake Tana- although none of these have formed huge radiations like the haplochromines have in Lake Malawi and Victoria. Overall, I think the work will lead to further “replications” in other lacustrine systems, facilitating the testing of the generality of the findings.

We are glad to hear reviewer 3 commends our efforts and shares our future outlook on how our study may lead to replication in other lakes and facilitate testing the generality of our key findings.

F. Suggested improvements: experiments, data for possible revision

Comment: (i) *The taxonomic treatment of the cyprinids should be updated to reflect current classifications. Rastrineobola is now in the Danionidae, but Cypriniformes (order) or Cyprinoidea (Superfamily) can still be used to unite this with taxa currently places in Cyprinidae.*

Response: We thank the reviewer for pointing this out and we have now updated the family from Cyprinidae to **Cyprinoidea** throughout our manuscript.

Comment: *Barbus (Table 1) is no longer applied to African taxa which are now Enteromius (as in Supplementary Table 1) and Labeobarbus.*

Response: we have adopted the suggestion by the reviewer and updated *Barbus* to “*Enteromius*” and also included “*Labeobarbus*”

Comment: (ii) *I am glad to see Astatotilapia nubila referred to as the “ancestor” of the haplochromine radiation and urge the authors to resist pressure from other reviewers/editors to try to shoehorn it into cladistic terminology which probably doesn't reflect reality!*

Response: We thank the reviewer for appreciating our approach to dealing with taxonomy in a complex evolutionary radiation.

Comment: (iii) *It is slightly confusing that the numbering of the cores goes from shallowest to deepest in the order 3, 2,1, 4. It might be easier to follow if it went 1,2,3,4 or 4,3,2,1.*

Response: We agree with the reviewer that the numbering of sites can be slightly confusing. However, the numbers cannot be changed at this stage because the sites were numbered during the field expedition in 2018 and the numbering is used in many other papers based on these cores. To ensure consistency, we have to maintain the numbering even though we agree that the scheme was an unfortunate choice.

Comment: (iv) *Supplementary Table 1: Oreochromini is a tribe of cichlid fishes (alongside haplochromini) and not a separate family from Cichlidae and Oreochromis variabilis should be under Oreochromini not Mastacembelidae.*

Response: We thank the reviewer for spotting these errors. The supplementary Table 1 is now corrected.

Comment: (v) *Figure 1 is generally nice, but the details of all the images of tiny fish in the lake are presumably meant to show lots of cichlids in deep water and a diversity of lineages in the shallows: it might work better to use fewer, bigger images- and perhaps omit the left hand side of the image.*

Response: We thank the reviewer for a great suggestion and we have now edited the figure to omit the left side.

We have thoroughly revised the manuscript to incorporate the above suggestions. Additionally, we have made adjustments to improve the clarity and structure of the paper.

Attached, please find the revised manuscript with the changes highlighted for your convenience.

We believe that the revised version of the manuscript now adequately addresses the concerns raised by the reviewers. We are very grateful to the reviewers for the many excellent suggestions that improved our manuscript. We hope the revised version will be deemed suitable for publication in *Nature*.

Thank you once again for your valuable guidance throughout this process. We look forward to hearing from you regarding the outcome of the revision.

Sincerely,

Nare Ngoepe and Ole Seehausen
on behalf of all co-authors

References

- Blaauw, M. and Andrés Christen, J. (2011) ‘Flexible paleoclimate age-depth models using an autoregressive gamma process’, *Bayesian Analysis*, 6(3), pp. 457–474.
- Bouton, N., Seehausen, O. and Van Alphen, J.J.M. (1997) ‘Resource partitioning among rock-

dwelling haplochromines (Pisces: Cichlidae) from Lake Victoria', *Ecology of freshwater fish*, 6(4), pp. 225–240.

Corbet, P.S. (1960) 'Breeding sites of non-cichlid fishes in Lake Victoria', *Nature*, 187, pp. 616–617.

Day, J.J. and Wilkinson, M. (2006) 'On the origin of the Synodontis catfish species flock from Lake Tanganyika', *Biology letters*, 2(4), pp. 548–552.

Fryer, G. and Iles, T.D. (1972) *The cichlid fishes of the Great Lakes of Africa: Their Biology and Evolution*. Oliver and Boy & London.

Graham, M. (1929) *The Victoria Nyanza and its Fisheries: A Report on the Fishing Survey of Lake Victoria, 1927-1928, and Appendices*. Crown Agents for the colonies.

Johnson, T.C. *et al.* (1996) 'Late Pleistocene Desiccation of Lake Victoria and Rapid Evolution of Cichlid Fishes', *Science*, 273(5278), pp. 1091–1093.

Liem, K.F. and Osse, J.W.M. (1975) 'Biological versatility, evolution, and food resource exploitation in African cichlid fishes', *American zoologist*, 15(2), pp. 427–454.

Lowe-McConnell, R.H. (1987) *Ecological Studies in Tropical Fish Communities*. Cambridge University Press.

Malinsky, M. *et al.* (2015) 'Genomic islands of speciation separate cichlid ecomorphs in an East African crater lake', *Science*, 350(6267), pp. 1493–1498.

McGee, M.D. *et al.* (2020) 'The ecological and genomic basis of explosive adaptive radiation', *Nature*, 586(7827), pp. 75–79.

Meier, J.I. *et al.* (2017) 'Ancient hybridization fuels rapid cichlid fish adaptive radiations', *Nature communications*, 8, p. 14363.

Moser, F.N. *et al.* (2018) 'The onset of ecological diversification 50 years after colonization of a crater lake by haplochromine cichlid fishes', *Proceedings of the royal society* [Preprint]. Available at: <https://royalsocietypublishing.org/doi/abs/10.1098/rspb.2018.0171>.

Njaya, F. *et al.* (2011) 'The natural history and fisheries ecology of Lake Chilwa, southern Malawi', *Journal of Great Lakes research*, 37, pp. 15–25.

Peel, R.A. *et al.* (2019) 'Species succession and the development of a lacustrine fish community in an ephemeral lake', *Journal of fish biology*, 95(3), pp. 855–869.

Sato, T. (1986) 'A brood parasitic catfish of mouthbrooding cichlid fishes in Lake Tanganyika', *Nature*, 323(6083), pp. 58–59.

Szidat, S. *et al.* (2014) '14C Analysis and Sample Preparation at the New Bern Laboratory for the Analysis of Radiocarbon with AMS (LARA)', *Radiocarbon*, 56(2), pp. 561–566.

Temoltzin-Loranca, Y. *et al.* (2023) 'A chronologically reliable record of 17,000 years of biomass burning in the Lake Victoria area', *Quaternary science reviews*, 301, p. 107915.

Tryon, C.A. *et al.* (2016) 'The Pleistocene prehistory of the Lake Victoria basin', *Quaternary international: the journal of the International Union for Quaternary Research*, 404, pp. 100–114.

Witte, F., De Winter, W. and Van Densen, W.L.T. (1995) 'Appendix II. Biology of the major fish species of Lake Victoria', *Fish stocks and fisheries of Lake Victoria-A handbook for field observations*.

Reviewer Reports on the First Revision:

Referees' comments:

Referee #1 (Remarks to the Author):

I previously thought this was a fine paper and I continue to think so. When I started to read this version, I was still a bit skeptical of the ecological versatility idea, in that it seemed more like a post hoc explanation than an idea that was explicitly tested in this study. I still harbor that feeling a bit, but the material in lines 260-275 make a pretty good case.

Two minor points:

24-26: "Since these habitat gradients are known to have played a major role in speciation, our findings suggest ecological versatility." This really doesn't follow: just because habitat gradients play a role in speciation doesn't say anything about the ecological versatility of taxa...or at least the reasoning is very implicit and not clear.

30: many consider the pace of diversification to not be important in identifying adaptive radiations

Referee #2 (Remarks to the Author):

The authors have fully addressed my comments from the previous review. This an excellent study that I would be pleased to see accepted and published. I have only a few minor suggested changes to the text, as detailed below.

Line 12 – suggest "which lineages radiate."

Line 22 – suggest "therefore there is no support"

Line 24 – suggest "open water habitats"

Line 36 – suggest "and the new availability of an abundance"

Line 42 – suggest "All these factors open up resources that were"

Line 45 – suggest "The ability of colonizing species to be able to access novel resources depends on a species' ecological versatility and timing of arrival, relative to other species."

Line 133 – suggest "of cyprinoid fossils"

Line 142 – suggest "cyprinoid fish"

Line 153 – *Typha* in italics (twice)

Line 200 suggest "pelagic cyprinoid"

Line 300 suggest "zooplanktivorous cyprinoid"

Line 321 suggest "islands in the sea"

Line 380 suggest "cyprinoid pharyngeal"

Referee #3 (Remarks to the Author):

Responses to reviewer 1 are somewhat lengthy, but seem well justified. 'The ecological flexibility' hypothesis is necessarily a bit vague (and has taken a bit of a beating in a recent paper!
<https://pubmed.ncbi.nlm.nih.gov/37326103/>)

but it remains in play and seems to be supported by the evidence presented that the cichlids were present in novel deep-water habitats from early on in the fossil record presented here. I think the arguments against them having invaded from elsewhere are well argued and I don't see any reason to change this. The more detailed suggestions of reviewers 2 & 3 all seem to have been implemented well.

Author Rebuttals to First Revision:

Reviewer 1:

I previously thought this was a fine paper and I continue to think so. When I started to read this version, I was still a bit skeptical of the ecological versatility idea, in that it seemed more like a post hoc explanation than an idea that was explicitly tested in this study. I still harbor that feeling a bit, but the material in lines 260-275 make a pretty good case.

We thank reviewer1 and appreciate the acknowledgment of our arguments for the ecological versatility hypothesis.

Two minor points:

Comment: 24-26: *“Since these habitat gradients are known to have played a major role in speciation, our findings suggest ecological versatility.” This really doesn’t follow: just because habitat gradients play a role in speciation doesn’t say anything about the ecological versatility of taxa...or at least the reasoning is very implicit and not clear.*

Response: We agree and the evidence for versatility in our paper is of course independent of the fact that other papers have shown that habitat gradients play a role in speciation. We have adjusted the text as follows: “Since these habitat gradients are **also** known to have played a major role in speciation, our **findings are consistent with the hypothesis that** ecological versatility was key to adaptive radiation, not priority by arrival order nor initial numerical dominance.” Line 25.

Comment: 30: *many consider the pace of diversification to not be important in identifying adaptive radiations*

Response: We have considered the reviewer’s suggestion and reworded line 31 “Adaptive radiation is the **rapid** diversification of a lineage into an array of species that possess traits enabling them to exploit different environments or resources”

Reviewer 2:

The authors have fully addressed my comments from the previous review. This an excellent study that I would be pleased to see accepted and published. I have only a few minor suggested changes to the text, as detailed below.

We thank reviewer2 for the positive feedback and for finding our study interesting and pleased for it to be accepted and published. We also thank them for their suggestions of minor changes which are very insightful. We have implemented all of them, with one exception, as per below.

Comment: Line 12 – suggest “which lineages radiate.”

Response: added on line 12: Ecological opportunity is thought to be a prerequisite for adaptive radiation, but little is known about the relative importance of a species’ ecological versatility versus effects of arrival order in determining **which** lineage radiates.”

Comment: Line 22 – suggest “therefore there is no support”

Response: Added on line 21: “There is no evidence of the radiating haplochromine cichlid lineage arriving before others, nor of their numerical dominance upon colonization, therefore **there is** no support for ecological priority effects.”

Comment: Line 24 – suggest “open water habitats”

Response: Added on line 23: “However, while many taxa colonized the lake early and several became abundant, only cichlids persisted in the novel deep and open **water** habitats once these emerged.”

Comment: Line 36 – suggest “and the new availability of an abundance”

Response: Added on line 37: “Ecological opportunity refers to the opening up of novel niches and **the new availability** of an abundance of resources.”

Comment: Line 42 – suggest “All these factors open up resources that were”

Response: reworded on line 43-44: “All these factors open up resources that **have in common the opening up of access to resources that** were previously inaccessible to the diversifying lineage.”

Comment: Line 45 – suggest “The ability of colonizing species to be able to access novel resources depends on a species’ ecological versatility and timing of arrival, relative to other species.”

Response: reworded on line 46-47: “**The ability of colonizing species to be able to access novel resources depends on a species’ ecological versatility and timing of arrival, relative to other species.**”

Comment: Line 133 – suggest “of cyprinoid fossils”

Response: changed on line 133: cypriniform to **cyprinoid**

Comment: Line 142 – suggest “cyprinoid fish”

Response: changed on line 142: cypriniform to **cyprinoid**

Comment: Line 153 – Typha in italics (twice)

Response: italicized “*Typha*” (twice) line 152

Comment: Line 200 suggest “pelagic cyprinoid”

Response: changed on line 199: cypriniform to **cyprinoid**

Comment: Line 300 suggest “zooplanktivorous cyprinoid”

Response: changed on line 305: cypriniform to **cyprinoid**

Comment: Line 321 suggest “islands in the sea”

Response: We did not understand this comment and did not change our text.

Comment: Line 380 suggest “cyprinoid pharyngeal”

Response: changed on line 374: cypriniform to **cyprinoid**

Reviewer 3:

Responses to reviewer 1 are somewhat lengthy, but seem well justified. ‘The ecological flexibility’ hypothesis is necessarily a bit vague (and has taken a bit of a beating in a recent paper! <https://pubmed.ncbi.nlm.nih.gov/37326103/>) but it remains in play and seems to be supported by the evidence presented that the cichlids were present in novel deep-water habitats from early on in the fossil record presented here. I think the arguments against them having invaded from elsewhere are well argued and I don't see any reason to change this. The more detailed suggestions of reviewers 2 & 3 all seem to have been implemented well.

We thank the reviewer3 for this positive feedback.